# Dynamically stable radiation pressure propulsion of flexible lightsails for interstellar exploration

Ramon Gao [1,2], Michael D. Kelzenberg [1,2] & Harry A. Atwater [1]✉

Meter-scale, submicron-thick lightsail spacecraft, propelled to relativistic velocities via photon pressure using high-power density laser radiation, offer a potentially new route to space exploration within and beyond the solar system, posing substantial challenges for materials science and engineering. We analyze the structural and photonic design of flexible lightsails by developing a mesh-based multiphysics simulator based on linear elastic theory. We observe spin-stabilized flexible lightsail shapes and designs that are immune to shape collapse during acceleration and exhibit beam-riding stability despite deformations caused by photon pressure and thermal expansion. Excitingly, nanophotonic lightsails based on planar silicon nitride membranes patterned with suitable optical metagratings exhibit both mechanically and dynamically stable propulsion along the pump laser axis. These advances suggest that laser-driven acceleration of membrane-like lightsails to the relativistic speeds needed to access interstellar distances is conceptually feasible, and that their fabrication could be achieved by scaling up modern microfabrication technology.

Centuries of astronomical observations and decades of robotic space exploration have been dedicated to the study of our own solar system. Exoplanets orbiting other sun-like stars were first conclusively detected in the 1990s[1], posing the question of whether life exists elsewhere in the universe[2]. However, exoplanets are far too distant to be directly imaged by telescopes, nor could conventional space probes reach them within the timescale of human civilization. Among the three space probes that have reached interstellar space, Voyager 1 has traveled the farthest, being presently 0.0025 light years away from our sun. The nearest star to our own is Proxima Centauri, 4.2 light years away, and hosts at least two exoplanets, with Proxima Centauri b lying in the habitable zone[3]. Exploration of such exoplanets will require significant advances in propulsion technology.

Lightsails utilize radiation pressure rather than reaction mass for spacecraft propulsion, potentially allowing space probes to reach far greater distances within a human lifetime. The concept dates to at least 400 years ago when Kepler observed that comet tails point away from

the sun as if blown by a solar wind[4,5]. Solar-powered lightsails have been demonstrated through the recent JAXA IKAROS[6], NASA NanoSail-D[7], and the Planetary Society LightSail missions[8], and have been proposed to enable a mission to the solar gravitational lens (SGL), nearly 0.01 light years from the sun, from which exoplanets could be studied with far greater resolution than with any conceivable telescope[9,10].

Whereas sunlight provides a relatively weak force for accelerating spacecraft in Earth's vicinity (~10 μN m$^{-2}$ for a perfect reflector at 1 AU), far greater accelerating forces can be produced if a high-power density laser is focused onto a lightsail. Laser-propelled lightsails can, in principle, be accelerated to relativistic velocities, offering a promising pathway for interstellar exploration using ultralight space probes for direct flyby missions[11–13]. Due in part to the announcement of the Breakthrough Starshot Initiative in 2016, which seeks to enable this capability within the next generation[14,15], recent investigations have explored the viability of laser-driven lightsails as a basis for interstellar spacecraft propulsion[13,16–18]. A major challenge for such lightsails is the

[1]Thomas J. Watson Laboratory of Applied Physics, California Institute of Technology, Pasadena, CA 91125, USA. [2]These authors contributed equally: Ramon Gao, Michael D. Kelzenberg. ✉e-mail: haa@caltech.edu

**Table 1 | Figures of merit for mechanical strength, including the stationary burst diameter $D_{SB}$ and maximum spin frequency $f_{max}$, of candidate lightsail materials**

| Material | Young's modulus, $E$ (GPa) | Tensile strength, $\sigma_T$ (GPa) | Density, $\rho$ (g cm$^{-3}$) | Thickness, $t$ for 0.1 g m$^{-2}$ (nm) | $D_{SB}$ burst for 67 Pa (m) | $f_{max}$ spin for 10 m$^2$ (Hz) |
|---|---|---|---|---|---|---|
| Silicon (111 surf.)[67] | 169 | 2.1 | 2.33 | 43 | 1.09 | 133 |
| Diamond (PECVD nano)[68] | 750 | Up to 7.5 | 3.27 | 31 | 1.35 | 179 |
| SiO$_2$ (tempered glass)[69,70] | 77 | Up to 1.0 | 2.42 | 42 | 0.52 | 91 |
| Si$_3$N$_4$ (LPCVD film)[44,71] | 270 | 6.4 | 2.7 | 37 | 3.96 | 215 |
| MoS$_2$ (multilayer)[41,72] | 200–330 | 21 | 5.02 | 20 | 19.9 | 285 |
| Aluminum | 72 | 0.50 | 2.80 | 36 | 0.16 | 58 |
| Polyimide[73] | 2.5 | 0.069 | 1.42 | 70 | 0.09 | 30 |

This table is not intended to suggest that lightsails should be constructed from uniformly thick, continuous membranes or to impose an upper limit for thickness, but rather to facilitate first-order comparison of the limiting structural capabilities of the candidate materials. More detailed properties are provided in Supplementary Table 1.

need to maximize reflectance while minimizing weight and limiting optical absorption to extremely low values, prompting multilayer or nanophotonic designs[19–23]. The lightsails must also be designed to be structurally and dynamically stable during acceleration, passively following the optical axis of propulsion[24–38] without collapsing or tearing. Many designs for rigid-body beam-riding lightsails have been proposed, and membrane deformation was modeled for gas-filled spherical lightsails[39], but to date, no study has considered the mechanical flexibility of meter-scale, fully unsupported membranes and its effects on acceleration stability. Notably, to achieve relativistic velocities, the Starshot mission calls for a ~10 m$^2$, ~1 g lightsail, which therefore, must be on the order of 100 atomic layers thick on average, including all framing or stiffening, suggesting that mechanical flexibility cannot be neglected in lightsail design.

Here, we examine the selection of materials, the structural and photonic design, and dynamic mechanical stability of flexible lightsail membranes to investigate whether interstellar lightsail spacecraft can be realized with real materials of finite stiffness and strength. We identify key material properties required for relativistic flexible lightsails, then develop a multiphysics numerical simulation approach to explore the deformation and passively stabilized acceleration of spinning flexible lightsails with either specular scattering concave shapes or flat membranes with embedded metagrating nanophotonic elements.

## Results
### Materials considerations
The Breakthrough Starshot Initiative[14] has challenged a global community of scientists and engineers to design a ~1-g interstellar probe that will travel 4.2 light years to reach Proxima Centauri b, the nearest known habitable-zone exoplanet, within ~20 years of launch, as well as the necessary propulsion, communication, and instrumentation systems for such a mission. To accelerate the spacecraft to ~0.2$c$, a ~10 m$^2$ lightsail weighing ~1 g would be propelled from low-earth orbit by an earth-based laser at incident power densities approaching ~10 GW m$^{-2}$, experiencing ~10,000 Gs of acceleration for ~1000 s[13,15]. A lightsail suitable for this mission must address immense engineering obstacles, challenging the limits of materials science and engineering.

One challenge is that the lightsail must have reasonably high optical reflectance to produce thrust from the accelerating beam yet must exhibit near-zero optical absorption ($\lesssim 1$ ppm) and high thermal emissivity to prevent overheating. A handful of dielectric and semiconductor materials have been identified as potentially viable candidates[16–18]. Optimized nanophotonic metamaterials comprising

combinations of such materials can offer favorable combinations of enhanced reflectance, low absorption, and high emissivity[19–22,40]. Lightsail materials and designs must also offer adequate mechanical strength and stiffness to endure the acceleration conditions necessary for interstellar propulsion. Table 1 shows key room-temperature mechanical properties and performance metrics for several candidate lightsail materials, with more details listed in Supplementary Table 1.

Bulk crystalline dielectrics and semiconductors such as Si, quartz (SiO$_2$), and diamond are hard, brittle, and have among the highest moduli and theoretical strengths of known bulk materials. Despite this, such materials are rarely used in bulk structural applications and are notorious for brittle failure in tension due to cracks initiated at surface defects. In practice, attainable specimen strength is limited almost entirely by the ability to fabricate device structures with defect-free surfaces. Although each of these materials can achieve remarkable degrees of purity and scale of manufacture, present-day technology has yet to produce pure, defect-free, submicron-thick membranes over 10 m$^2$ areas.

Among two-dimensional crystals, MoS$_2$ appears particularly promising for lightsail applications owing to its high strength and refractive index[20]. The reported tensile strength for micron-scale suspended membranes of mono- or bi-layer MoS$_2$ is nearly three times higher[41] than that of any other material listed in Supplementary Table 1. Understanding the achievable strength and optical transparency of MoS$_2$ films fabricated over large or nonplanar surfaces at relevant layer thicknesses and elevated temperatures is of considerable interest.

Another interesting class of materials includes amorphous or nanocrystalline deposited thin films. Promisingly, submicron-thick silicon nitride membranes have been fabricated at wafer scale and further patterned with photonic crystal designs for near-unity reflectance[42,43]. Widely employed in MEMS and cavity optomechanics applications[44,45], high-stress silicon nitride (Si$_3$N$_4$) is a particularly promising candidate material for lightsail development due to its ultralow extinction coefficients on the order of 10$^{-6}$ at near-infrared wavelengths and large modulus and tensile strength.

Ultimately, considerable effort will be required to develop any suitable materials system(s) to the scale of manufacture required for the interstellar lightsails proposed by the Starshot initiative, and careful consideration must be paid to the resulting mechanical and optical properties of the lightsail materials over a wide range of operating temperatures to endure the forces and optical intensities of the propulsion laser beam.

## Stability considerations

Our work addresses two key challenges for stable lightsail acceleration and potential solutions to them (Fig. 1): *beam-riding stability*, the ability of the lightsail to follow along the beam axis without external guidance, and *structural stability*, the ability of the lightsail to survive the acceleration sequence without collapsing, rupturing or excessively deforming.

Passive beam-riding stability is necessary for relativistic lightsails because the large acceleration distance precludes closed-loop beam control to provide trajectory corrections. When the lightsail becomes misaligned with the beam, its design must produce corrective optical forces based on spatial power density variations on the lightsail. In practice, the laser system would be constructed no larger nor more powerful than necessary to achieve the desired final velocity, such that the system would operate at or near the diffraction limit during the final acceleration phase. As depicted in Fig. 1a, a flat specularly reflective disk does not offer beam-riding stability and will tilt and veer away from the beam, whereas certain geometrically concave reflector shapes, including cones[26-28], hyperboloids[24], paraboloids, and other parametric shapes[37] can achieve stable beam-riding behavior. Convex shapes such as spheres can exhibit stability using more complex higher-order beam profiles[26]. Nonspecular surfaces can be employed to produce restoring forces and torques, even for flat lightsails[23,29-33,38], and have been shown to enhance lateral and rotational maneuverability of solar lightsails[46].

Our study addresses marginal (undamped) beam-riding stability during acceleration, where the lightsail exhibits bounded, oscillatory displacement and tilting about the beam axis in response to an initial beam-lightsail misalignment. Continuous perturbations to the beam-lightsail alignment could cause the oscillatory motion to grow in amplitude, eventually ejecting marginally stable lightsails from the beam. Furthermore, for nonrigid structures such as flexible membranes[47], gradual energy buildup in vibrational or acoustic modes could also destabilize or overstress the lightsail. Therefore, interstellar lightsails will likely require either active or passive means of damping their beam-riding oscillations and structural vibrations to achieve asymptotically stable propulsion. Passive damping could be achieved with damped internal degrees of freedom[36], nonlinear optical materials[35], or materials with highly varying temperature-dependent optical properties to enable hysteresis of the restoring forces.

Regarding structural stability, the lightsail must survive the acceleration forces without experiencing mechanical failure or deformations that would disrupt beam-riding. Optimized membrane designs could incorporate multiple materials[21], complex geometries, and intricate spatial patterning, e.g., perforations[20,22,35,43] or optical resonators[23,29,30,38,40] to maximize reflectance, emissivity, and tensile strength. Rigid shell theory has been applied to study stress distributions in parametrically shaped lightsails[48], and 2D analytic and finite-element models of deformation instabilities have been reported for uniformly illuminated lightsails[49], but the behavior of unsupported or loosely supported flexible membranes subject to nonuniform forces is generally complex[47].

Thin unsupported membranes will generally collapse and crumple upon themselves when subject to focused laser propulsion (Fig. 1b, left). A curved surface offers greater structural rigidity than a flat membrane while also conferring the benefits of improved stress distribution that make thin curved shells useful in structural applications. However, open concave shapes such as cones and paraboloids are still prone to collapsing by elongation (Fig. 1b, center left). When curved lightsails become slightly deformed, their elongated regions present larger cross-sectional areas and smaller incidence angles, resulting in an increased effective photon pressure, whereas the narrowing regions similarly experience decreasing effective photon pressure, furthering the distortion and leading to collapse. Adding structural reinforcement or framing to resist distortion comes with a mass penalty.

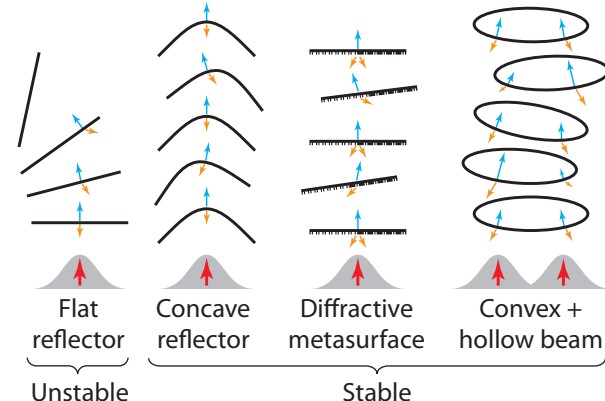

(a) Beam-riding stability

Flat reflector — Concave reflector — Diffractive metasurface — Convex + hollow beam

Unstable — Stable

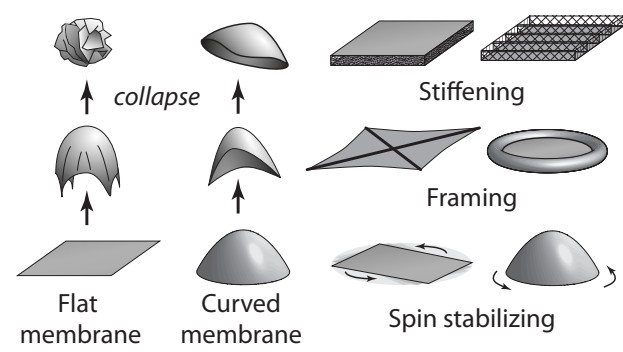

(b) Structural stability

*collapse* — Stiffening — Framing

Flat membrane — Curved membrane — Spin stabilizing

**Fig. 1 | Conceptual illustrations of design approaches.** Designs for achieving **a** beam-riding stability, and **b** structural stability, in lightsail membranes. In panel **a**, the red arrows depict the accelerating beam position, the orange arrows indicate the direction of reflected light, and the blue arrows indicate the force of radiation pressure.

Potential structural reinforcement approaches include microlattices[50], gas-filled envelopes[39,51], annular tensioning, fractal supports[52], tensegrity structures[53], or lamination with low-density or corrugated backing layer(s). Ultimately, given mass and material constraints, even a structurally rigidified lightsail will likely deform during acceleration, potentially changing the membrane's stress distribution or altering its beam-riding properties. The proposed lightsail membranes are generally partially transparent; thus, any frame or backing materials may still be exposed to high laser intensities even if placed behind the lightsail surface, further limiting materials selection.

Alternatively, spin-stabilization can prevent shape collapse by effectively rigidifying the lightsail via inertial tensioning and gyroscopically stabilizing the lightsail to resist tilting, all while avoiding the added mass and complexity of structural reinforcement. However, spin-stabilization greatly complicates the dynamics of the lightsail, particularly for flexible membranes prone to complex instabilities[47], and is not necessarily effective for all structures under all conditions. Perhaps most counterintuitively, gyroscopic effects can disrupt the beam-riding behavior of certain lightsail designs that would be dynamically stable under nonspinning (rigid-body) conditions, particularly in the case of angular misalignment between the beam and spin axes[26,27]. Thus, the use of spin-stabilization for ultrathin flexible lightsails can be a challenging design objective.

Table 1 introduces two figures of merit to facilitate first-order comparison of the limiting structural capabilities and to provide insights into the general viability of constructing large-area

structurally stable lightsails from the candidate materials. The *stationary burst diameter* $D_{SB}$ is the maximal diameter at which a flat circular membrane (rather than a plate or shell) of areal density $0.1\,\mathrm{g\,m^{-2}}$, rigidly clamped at its perimeter, can sustain a pressure of $67\,\mathrm{Pa}$ (the effective photon pressure of $10\,\mathrm{GW\,m^{-2}}$ illumination for unity reflectance) applied to one side without rupturing[54]. This pertains to the construction of a perimeter-supported lightsail, e.g., spanning a ring-shaped support frame, but rather than making specific assumptions about the mass, rigidity, or pretensioning of such a support structure, we consider the simpler and more conservative case in which the perimeter is stationary and rigidly clamped without pretensioning. Precluding free flight of the membrane, $D_{SB}$ should thus be interpreted as a comparative figure of merit rather than a practical size limit for perimeter-supported interstellar lightsails. Realistically, the support structure must have finite (preferably small) mass so that it could be accelerated with the lightsail, and the beam would necessarily taper off at the lightsail edge, enabling larger lightsails to be constructed than indicated by $D_{SB}$ (see Supplementary Note 2 for example cases). The design of a practical lightsail spacecraft must address its specific support structure(s) and payload(s) and must also consider optical and mechanical properties of the membrane throughout the range of illumination conditions and operating temperatures experienced during acceleration—none of which are captured by $D_{SB}$, although we address some of these issues in greater detail in our numerical simulations below. But interestingly, some candidate membrane materials ($Si_3N_4$, $MoS_2$) are, in principle, strong enough to span $10\,\mathrm{m^2}$ areas ($D_{SB} > 3.6\,\mathrm{m}$) with perimeter support—even in the stationary case. On the other hand, conventional solar lightsail materials such as aluminum and polyimide are considerably weaker and likely unable to span meter-scale areas between structural supports in interstellar lightsail applications, noting that they can more decisively be ruled out on the basis of their optical absorption alone[16].

The second figure of merit is the *maximum spin frequency $f_{max}$* at which a flat, $10\,\mathrm{m^2}$ circular membrane ($D = 3.6\,\mathrm{m}$) could be spin-stabilized without rupturing due to tensile failure[55]. For the designs considered here, relatively high spin frequencies are required to produce both shape stability and beam-riding stability, with resulting stress approaching a significant fraction of the materials' tensile limits. The viability of spin stabilization depends on the spin frequency, the acceleration conditions, and the specific design of the lightsail.

## Mesh-based multiphysics modeling of flexible lightsails

To simulate flexible lightsail membranes of various shapes and optical designs, a triangular surface mesh is constructed (Fig. 2a) to represent the membrane as a mass-spring dynamical model in finite-difference time-domain simulations. Light–matter interactions are calculated over the enclosed triangular mesh elements: Incident light produces photon pressure forces, optical absorption heats the lightsail, and thermal radiation cools the lightsail (Fig. 2b). In the simplest type of optical interaction, the photon pressure force results from specular reflection (Fig. 2c), with the resulting force directed normal to the surface. We first assume fixed values for reflectance and absorptance to model curved and flat specular lightsails. Then, we improve the specular surface model to include angle-dependent reflectance and absorptance based on Fresnel coefficients and consider the effects of multiple reflections of light within concave curved lightsails using simplified ray tracing. Finally, we present simulations of nonspecularly reflecting surfaces, demonstrating that diffractive metagratings (Fig. 2d) allow flat lightsails to achieve beam-riding stability. With future work, lightsails made from optical metasurfaces (Fig. 2e) with various optical behavior could be studied using this simulation approach.

Our simulations have studied only the first few seconds of acceleration, with the lightsail being misaligned to the beam at $t = 0\,\mathrm{s}$, to observe its shape evolution and whether its motion exhibits marginal

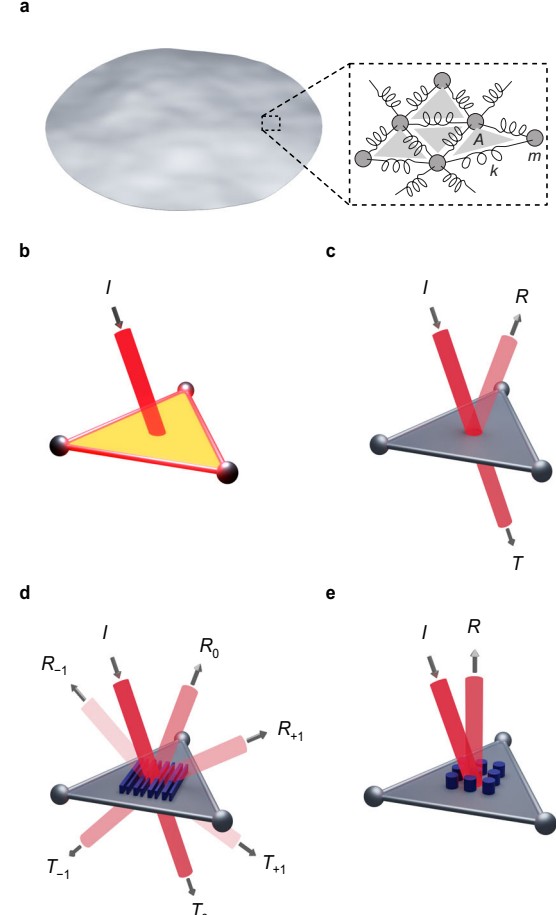

**Fig. 2 | Modeling flexible lightsails and light–matter interaction with a mesh-based time-domain simulator. a** Ultrathin and meter-scale lightsails and their deformations can be modeled by a mesh comprising masses $m$ (nodes) connected by springs with stiffnesses $k$ (edges), enclosing triangles of area $A$. Light–matter interactions are calculated for each mesh triangle based on discretization of the incident light as localized beam $I$. Modeled behaviors include **b** absorption of light and thermal emission, heating and cooling the structure, driving heat flow, thermal expansion, and changes in material properties; **c** specular reflection $R$ and transmission $T$ of light, producing photon pressure, and in some cases, causing reflected light to impinge other triangles; **d** optical diffraction with exemplary reflected orders $R_{\pm 1}$, $R_0$ and transmitted orders $T_{\pm 1}$, $T_0$ from periodic wavelength-scale surface patterning, producing transverse directional forces from photon pressure, and **e** optical wavefront shaping such as reflective beam steering with subwavelength optical metasurfaces.

stability over many periods of beam-riding oscillation, to determine temperature and stress distributions and to identify thermal or mechanical membrane failure. Future efforts could also consider the effects of local temperature and strain on optical and mechanical properties, study damping or active control surfaces[56], include beam profiles varying in time or distance from the source, or address relativistic effects necessary to model the full acceleration duration to interstellar velocities.

## Dynamics of flexible curved lightsails

The simulated behavior of flat versus curved (paraboloid) lightsails and the effects of spin stabilization are depicted in Fig. 3, using optical and mechanical properties corresponding to a ~43 nm-thick Si membrane ($0.1\,\mathrm{g\,m^{-2}}$). We assume fixed values for specular reflectance (0.45), absorption ($1.4 \times 10^{-7}$), and emissivity (0.1), with the latter two values being significantly higher than expected for the Si membrane alone. This is done to approximate a radiative cooling surface of the

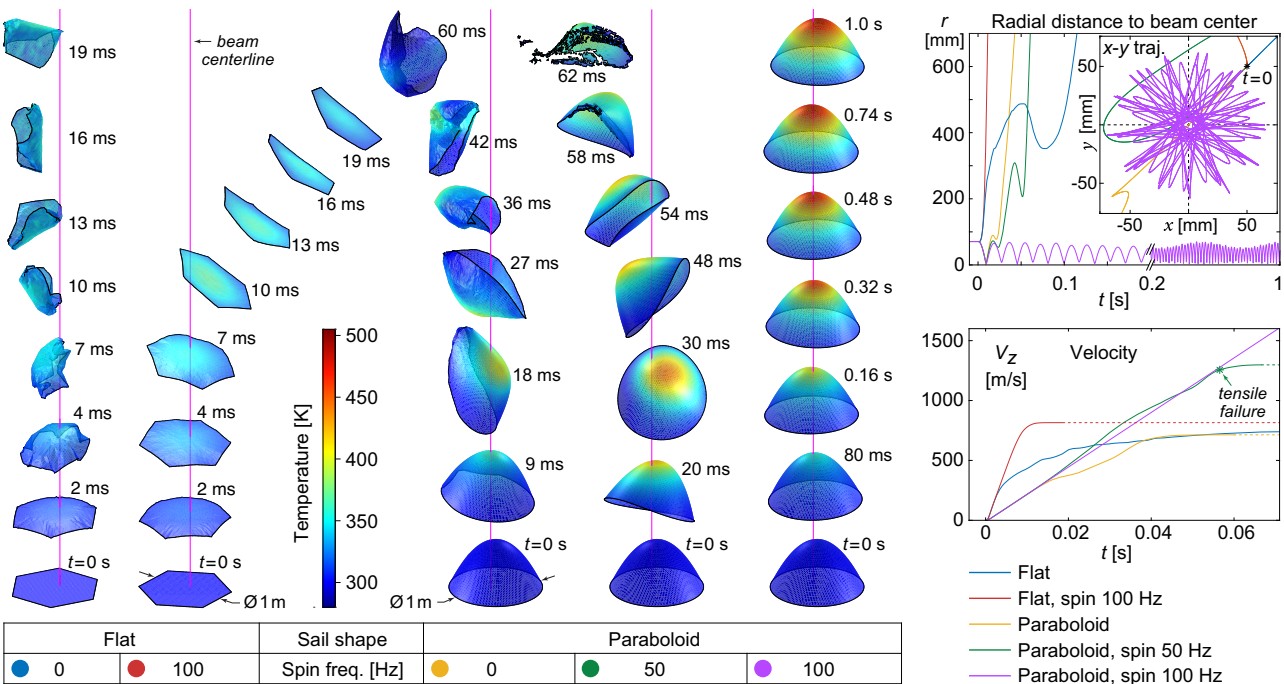

**Fig. 3 | Simulation results for flat versus curved specular silicon lightsails, with and without spin stabilization.** All lightsails are 43 nm thick, 1 m in diameter, and initially offset by 71 mm from the center ($x_0 = y_0 = 50$ mm). Illumination is in the $+z$ direction, with a Gaussian profile ($I_0 = 5$ GW m$^{-2}$, $w = \sqrt{2} \times R_{sail} = 0.7$ m, $\lambda = 1.55$ μm) and ramps on over 1 ms starting at $t = 0$ s. *Left*: Surface renderings show the temperature, shape, and lateral position of each lightsail at the indicated times during the simulation. The vertical magenta lines show the beam centerlines. All lightsail images appear at the same size and temperature scale; however, their vertical positions have been shifted for presentation, the view angle for each individual simulation was chosen to allow display without overlapping, and surface shading was applied to enhance depiction of shape. *Right plots*: The distance between the lightsail center of mass and the beam centerline (above), and the lightsail $z$ velocity (below), plotted versus time. Animations of all five simulations are available as Supplementary Movie 1.

undetermined design required to avoid the risk of thermal runaway for the Si membrane[57]. We first consider a 1-m diameter flat lightsail (of hexagonal shape to better illustrate shape deformations), illuminated by a Gaussian beam ($\lambda = 1550$ nm, $I_0 = 5$ GW m$^{-2}$, $w = \sqrt{2} \times R_{sail} = 707$ mm), with the lightsail initially offset by $x_0 = y_0 = 50$ mm from the optical axis. This lightsail diameter serves as a compromise between available computational resources and modeling the membrane behavior with high mesh fidelity, while the initial lightsail-beam offset and the spin frequencies were chosen to present visually exemplary acceleration scenarios in Fig. 3 and do not indicate specific stability thresholds. Without spin-stabilization, the flat membrane is structurally unstable and collapses upon itself as expected. Spin-stabilization ($f_{spin} = 100$ Hz) prevents collapse, but lacking beam-riding stability, the flat lightsail quickly veers away from the beam axis. Keeping the same lightsail properties and illumination conditions, we next consider a paraboloid lightsail, whose shape could potentially offer beam-riding stability. However, without spin-stabilization, the flexible paraboloid membrane quickly becomes elongated and collapses. Spinning the lightsail at $f_{spin} = 50$ Hz delays but does not prevent collapse; the shape gradually distorts through elongation and beam-riding tilt oscillations until a periphery region eventually becomes reverse-illuminated, causing the edge to fold over and initiate tensile failure. With adequate spin stabilization ($f_{spin} = 100$ Hz), the shape remains stable, and beam-riding stability is achieved throughout the simulated acceleration duration (1 s). Animations of all five cases are included in Supplementary Movie 1.

All lightsails in Fig. 3 have approximately the same diameter and thus total incident power. The paraboloids accelerate more slowly than the flat lightsails as they have more surface area and thus mass. Furthermore, their sloped surfaces propel the lightsail along the $z$-direction less efficiently due to reduced momentum change for angled light reflections and part of the resulting photon pressure being directed radially. Therefore, beam-riding curved lightsails generally reduce acceleration compared to a flat membrane of the same composition. Looking at curved geometries more carefully, light reflected from these sloped areas could strike other parts of the lightsail, imparting additional force there, thus potentially affecting lightsail acceleration and stability. Therefore, we improved our simulation to consider multiple reflections within the lightsail using a ray-tracing approach (Supplementary Note 3) and also implemented angle-dependent reflectance and absorptance based on Fresnel coefficients, thus better modeling the optical behavior of curved lightsails.

Figure 4 compares the dynamics, shape, and temperature behavior of the 1-m diameter, 43-nm-thick 100 Hz spin-stabilized paraboloid Si lightsail from Fig. 3, under otherwise identical simulation conditions, with and without the effects of multiple reflections. This membrane is dynamically stable for incident beam interaction only, but owing to its modest reflectivity (0.45 for $\lambda = 1550$ nm at normal incidence), the stability is substantially disrupted by the secondary reflections. Secondary reflections do increase the total photon pressure on the lightsail, resulting in faster acceleration, but reflected light striking the opposite side of the lightsail induces additional forces and torques there, which counteract the restoring forces and torques produced by the first reflection (Fig. 4a), thus destabilizing the lightsail (Fig. 4b–f). The concentration of reflected light in certain regions also results in localized heating, with the peak temperature increasing from ~502 to ~645 K (Fig. 4d). Animations of these and other ray-tracing-based simulations are shown in Supplementary Movie 2.

Localized heating could be problematic for any curved lightsail whose geometry concentrates reflected light in certain areas. We have modeled absorptive heating, radiative cooling, thermal conduction, and radiative heat transfer (see Supplementary Note 4); for the simulated membranes, thermal conduction and radiative heat transfer are negligible, and temperatures depend mostly on emissivity and optical

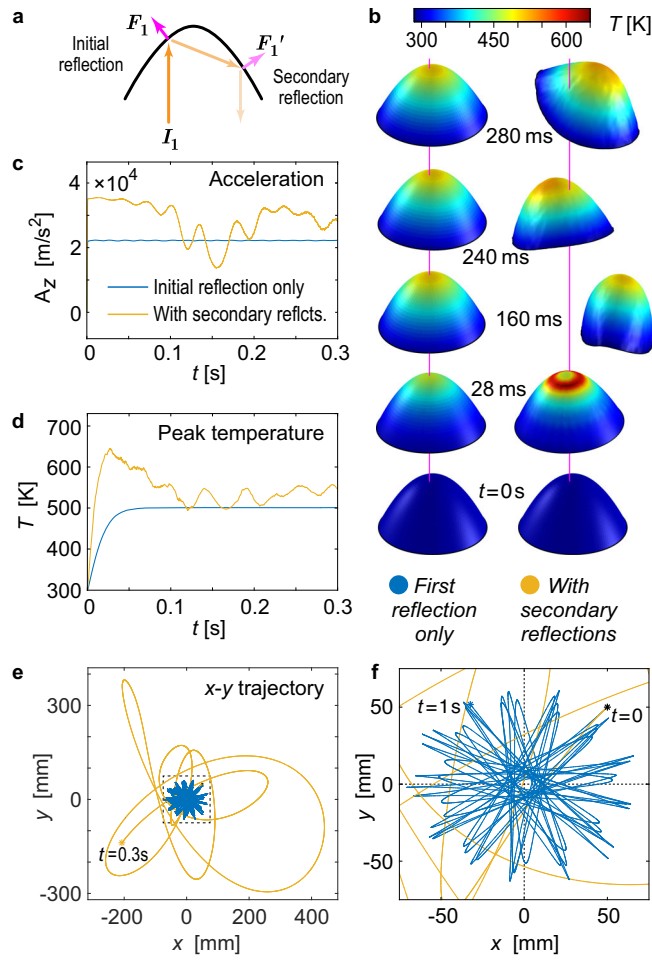

**Fig. 4 | Effects of secondary light reflections within a spin-stabilized flexible 43-nm thick paraboloid lightsail. a** Illustration of lightsail destabilization due to additional force $F_1'$ induced by secondary reflection on top of primary force $F_1$; **b** shape, lateral position, and temperature $T$ profile; **c** acceleration along $z$-axis; **d** peak temperature; and **e**, **f** trajectory of a specular paraboloid lightsail, with (yellow) and without (blue) the effects of secondary light reflections within the lightsail. The lightsail geometry, material, simulation conditions, and display settings are identical to the paraboloid lightsail from Fig. 3. Animations of these and other simulations based on ray tracing are available in Supplementary Movie 2.

absorption. Elevated temperatures can cause thin lightsail materials to weaken, melt or decompose via sublimation[20]. By limiting the mass loss to 1%, for a 1 g, 10 m² lightsail for 1000 s, for crystalline Si[58], we might predict a limiting temperature of ~1300 K. However, for semiconductors, free-carrier absorption increases dramatically with temperature, which may initiate destructive thermal runaway at a much lower threshold temperature. Furthermore, two-photon absorption may trigger thermal runaway above certain laser intensities, regardless of initial temperature. For a reported optimized Si-based nanophotonic lightsail[57], a runaway threshold of up to ~500 K and a maximum incident power intensity of ~5 GW m⁻² was predicted. This motivated our initial design and acceleration conditions for the Si paraboloid lightsails, but upon including focused secondary reflections, this particular design and illumination conditions would result in a thermal runaway (Supplementary Note 4).

Despite challenges associated with curved lightsails, other curved lightsail designs may be viable, and extracting optical thrust from secondary reflections can improve propulsion efficiency. For example, secondary reflections do not destabilize a Si₃N₄ paraboloid of identical shape and mass (owing to its lower reflectance) but improve its acceleration (Supplementary Note 5). Alternatively, shallower

spin-stabilized curved shapes can achieve beam-riding stability without encountering conditions producing secondary internal reflections[26]. For bulk crystals, curved profiles could expose planes that are weaker or more difficult to passivate, whereas for polycrystalline or 2D crystalline membranes, conforming to curved surfaces requires joints or grain boundaries, which may introduce weakness or absorption. For these reasons, we chose to investigate flat membranes as an alternative to curved shapes for the remainder of the study.

Despite the relatively high membrane reflectance and reasonable temperatures of Si lightsails (assuming adequate radiative cooling and absent focused secondary reflections), submicron surface contamination (e.g., due to dust impacts during acceleration) or by logical extension a localized defect or momentary local power excursion exciting two-photon absorption could initiate the propagation of catastrophic thermal runaway across the entire lightsail[59]. We therefore turn our attention to Si₃N₄, used extensively in other high-temperature applications, whose larger optical bandgap (~5 eV) and lower free-carrier absorption are attractive. Amorphous LPCVD Si₃N₄ films of excellent optical quality suggest an easier route for fabrication over large or complex surfaces[42,43]. A drawback of Si₃N₄ is its modest refractive index ($n$ ~ 2), resulting in lower reflectance and less efficient diffraction.

For Si₃N₄ lightsails, we estimate the temperature limit for vacuum decomposition (choosing 1% decomposition over 1000 s) to be ~1600 K[60]. Practical limits would likely be lower, as nitrogen evolution would leave Si-rich material with higher optical absorption, leading to thermal runaway. Other high-temperature risks include weakening, stress distribution changes, activation of traps or defects, or material crystallization. Further experimental measurements are needed to determine the limiting temperatures and power densities for Si₃N₄ lightsails.

## Optical design for passive stabilization of flat lightsails

Passive stabilization of lightsail dynamics requires restoring forces and torques. Concave curved shapes can achieve this via their shape alone, but flat specular lightsails lack beam-riding stability because specular reflection only produces forces normal to the surface. One approach to obtain beam-riding flat lightsails is to use engineered optical anisotropy in diffractive gratings based on nematic liquid crystals or asymmetric dielectric metagratings[29,38], where anisotropic scattering of incident light into grating diffraction orders manifests in optical forces transverse to the membrane. Moreover, optical metasurfaces comprising subwavelength scatterers[30,33,34] can shape the wavefronts of scattered light to redirect incident photon momentum in anomalous ways for beam-riding stability.

We present stable designs for flat lightsails spinning at 120 Hz by designing asymmetric diffractive Si₃N₄ metagratings using linearized stability analysis based on rigid-body Floquet theory[61,62] (Fig. 5a). Previously, we successfully fabricated and optically characterized similar metagratings patterned in silicon nitride membranes[38]. A specifically designed pair of mirror-symmetrically arranged metagratings can passively stabilize translations and rotations along one axis[29,38]. Consequently, we employ two perpendicularly arranged distinct metagratings to enable stabilization of translations along both $x$ and $y$ and rotations $\theta$ about $\mathbf{y}_{BF}$ (pitch) and $\phi$ about $\mathbf{x}_{BF}$ (roll) by introducing stabilizing forces and torques for their respective design planes and polarization (Fig. 5a). Studying the initial acceleration of our lightsail designs, subject to an initial alignment error, allows us to verify the predicted dynamical stability, and importantly, to investigate whether these spinning lightsails retain their beam-riding stability without the assumption of rigidity.

Asymmetry in the intensities of the $m = \pm 1$ diffracted orders provides the mechanism for lateral restoring forces, while asymmetry in the angular dependence of optical thrust can enable restoring torques. We calculate the normalized optical forces and torques induced on a

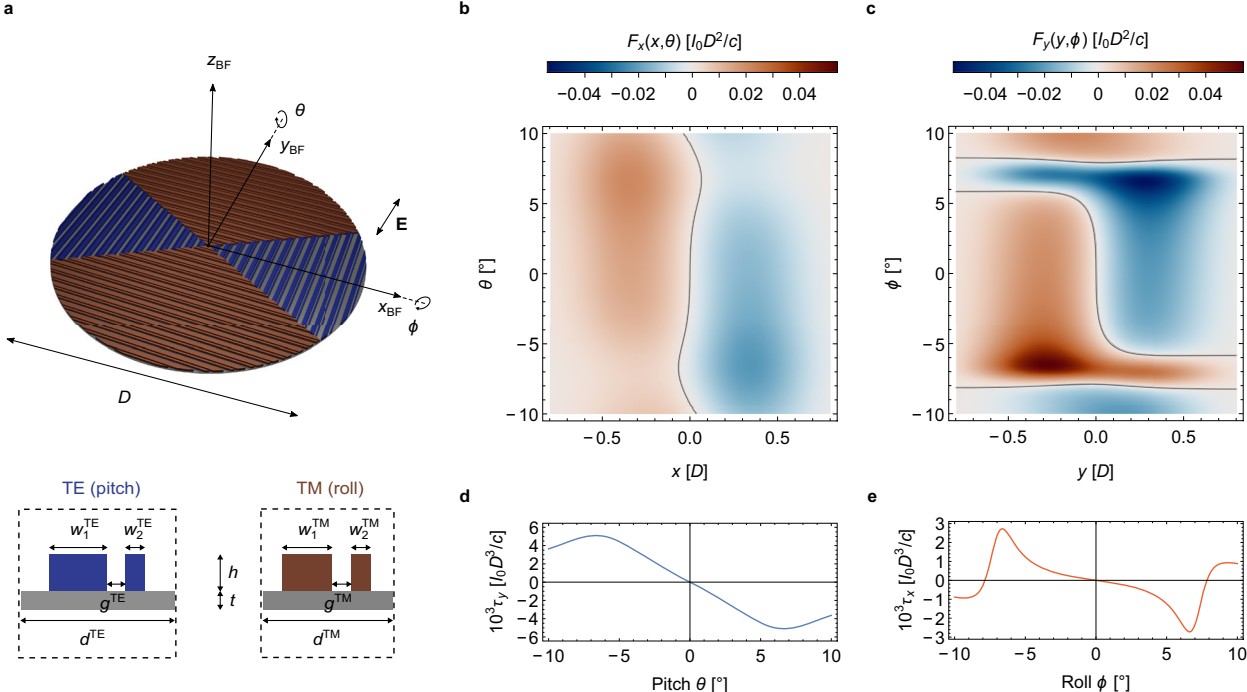

**Fig. 5 | Optical design with restoring forces and torques on a metagrating-based silicon nitride lightsail. a** Top: Conceptual illustration of lightsail patterning with two mirror-symmetrically arranged metagrating designs. Tilts $\theta$ and $\phi$ are defined as rotations about the body-frame axes $\mathbf{x}_{BF}$ and $\mathbf{y}_{BF}$ of the lightsail of diameter $D$, respectively, with the electric field vector $\mathbf{E}$ being aligned to $\mathbf{y}_{BF}$. Bottom: Unit cells of metagrating design operating for transverse-electric (TE) polarization (left) and transverse-magnetic (TM) polarization (right), with geometrical parameters listed in the "Methods" section. **c** Normalized optical force $F_x$ as a function of normalized translation $x$ and tilt $\theta$, being self-stabilizing due to $dF_x(x, \theta = 0)/dx < 0$ between $\pm 0.35D$. Contour line depicts the equilibrium position, $F_x = 0$. **d** Normalized optical force $F_y$ as a function of normalized translation $y$ and angle $\phi$ being self-stabilizing due to $dF_y(y, \phi = 0)/dy < 0$ between $\pm 0.3D$. Contour lines depict $F_y = 0$. **e** Optically induced torques $\tau_y(\theta)$ and $\tau_x(\phi)$, evaluated at the beam center. Self-restoring behavior ($d\tau_y(\theta)/d\theta < 0$, $d\tau_x(\phi)/d\phi < 0$) occurs over the approximate range of $\pm 6.5°$ in both directions. In **b–e**, $I_0$ denotes the incident beam intensity, while $c$ is the speed of light.

rigid lightsail of the proposed design over a range of incidence angles ($\theta$, $\phi$) and translational offsets ($x$, $y$), which are independent of acceleration distance $z$ and yaw tilt $\psi$ because we neglect beam divergence and assume synchronized rotation of the polarization. Perturbations to the beam-lightsail angular velocity alignment could potentially be addressed by a restoring yaw torque, which would be introduced by rotating the metagratings by an angle relative to the lightsail axis (Supplementary Fig. 9). Stabilizing behavior is evident from the negative slopes of $F_x$ and $F_y$ versus $x$ and $y$, respectively, with zero crossings (equilibrium positions; gray isolines) present near the beam center ($x$, $y = 0$) over the full $\pm 10°$ range of pitch angles $\theta$ and over a ~$\pm 5°$ range of roll angles $\phi$, respectively (Fig. 5b, c). The relative insensitivity of lateral equilibrium position to the tilt angle appears to benefit stability in the spinning case.

Restoring torques limit angular rotation relative to the optical axis, although the situation is less straightforward for the spinning case. Beam-center optical torques about $x$ and $y$ exhibit stabilizing polarity and derivative over a $\pm 6.5°$ range of tilt angles (Fig. 5d, e). While the TE metagrating provides a larger torque about $y$, the TM metagrating yields slightly stronger optical forces along $y$. $\tau_x(\phi)$ appears markedly nonlinear for rotations beyond ~$\pm 1.5°$, giving rise to nonlinear dynamics and possibly resulting in coupling between distinct frequency components. Our time-domain numerical simulations allow this behavior to be studied by considering the full angle-resolved optical response of the metagratings.

### Dynamics of metagrating-based lightsail

To verify our predictions about the dynamical stability of rigid lightsails patterned with the reported composite metagrating design ($D = 1$ m, $m = 0.867$ g) and propelled by a Gaussian beam ($\lambda = 1064$ nm,

$I_0 = 1$ GW m$^{-2}$, $w = 0.4D = 40$ cm), we numerically solved its equations of motion. The multiphysics dynamics of corresponding flexible lightsails with the same metagrating motif and laser illumination were also simulated using our mesh-based modeling approach.

We present here an exemplary case of passive stabilization of a flexible metagrating lightsail with an initial translational offset of $x_0 = y_0 = 5$ cm in the lightsail position relative to the beam optical axis and an initial (pitch and roll) tilt of $\theta_0 = \phi_0 = -2°$ (Fig. 6). Snapshots of the flexible lightsail position, orientation and shape every 0.5 s are shown in Fig. 6a, with an animation of the simulation being available as Supplementary Movie 3. For the studied duration of $t = 5$ s, the lightsail oscillates about the beam axis while remaining relatively flat and level thanks to sufficiently large tensioning forces arising from spin-stabilization. However, on closer inspection, the nonuniform beam induces shape deformations (Supplementary Fig. 11), with a maximum displacement from a perfectly flat shape of ~3 mm (Supplementary Fig. 12a). Due to the finite absorptivity of Si$_3$N$_4$, the lightsail center reaches a maximum temperature of 959 K. In contrast, the peripheral area remains significantly cooler (Supplementary Fig. 13a), heating up to a maximum temperature of 489 K. The slower heating at the edge of the lightsail can be attributed to a combination of beam absorption and the heat capacity of Si$_3$N$_4$. The peak temperature is sufficiently below the estimated vacuum decomposition temperature of Si$_3$N$_4$[60], although this temperature is likely too hot for most payloads. We assumed a hemispherical emissivity of 0.1 for bifacial thermal radiation from thin Si$_3$N$_4$ membranes, which could be increased with additional metasurface designs for selective thermal radiation in the mid-infrared regime or the addition of other material layers[20,21,63]. Our simulation predicts a maximum strain of 0.091% in the Si$_3$N$_4$ membrane (Supplementary Fig. 14a), which translates to a tensile stress of ~246 MPa,

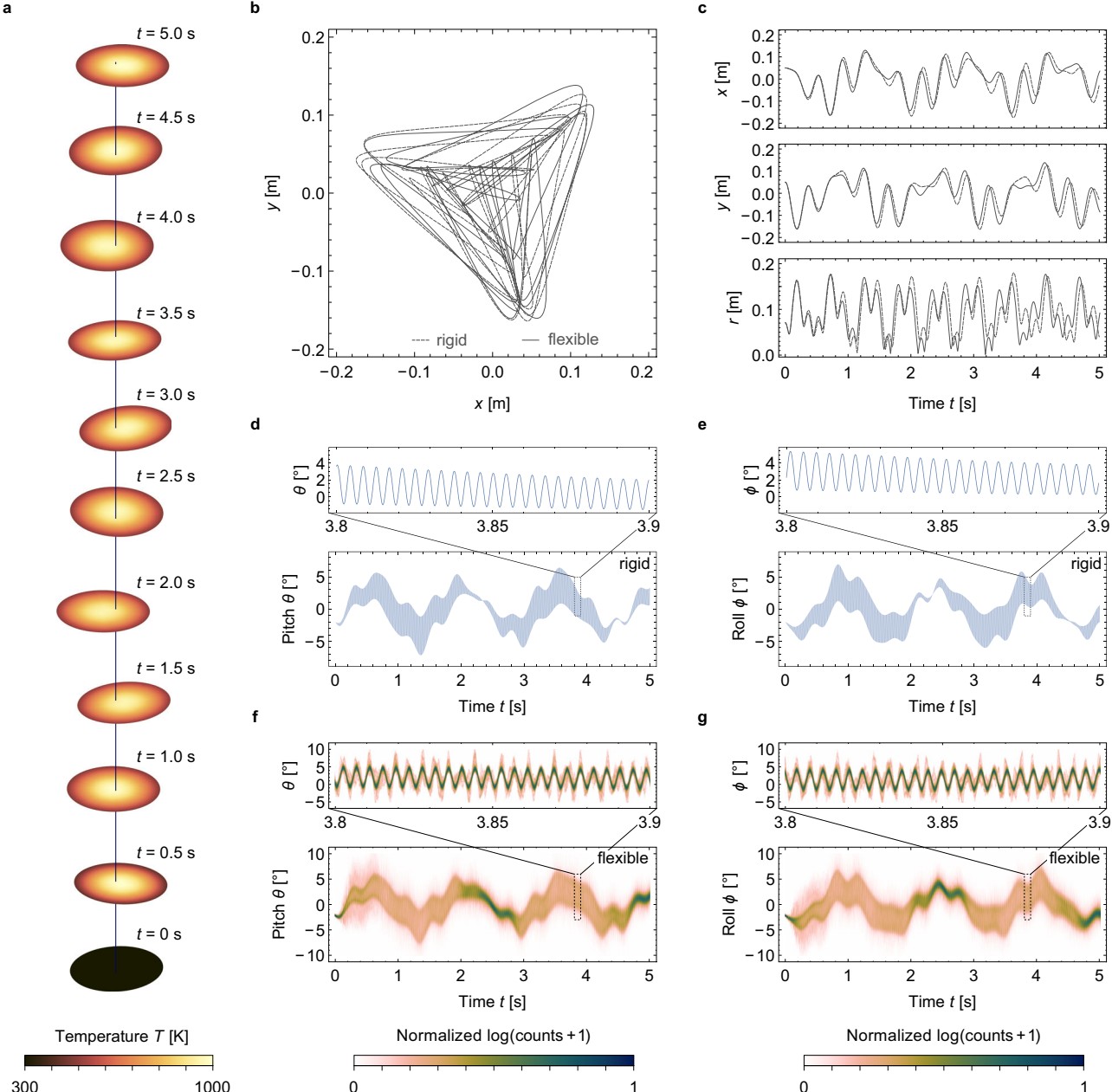

**Fig. 6 | Acceleration dynamics of a flexible and a rigid spinning $Si_3N_4$ lightsail based on the reported composite metagrating pattern.** Lightsails ($D = 1\,\text{m}$, $t_{\text{eff}}^{\text{TE}} = 400\,\text{nm}$, $t_{\text{eff}}^{\text{TM}} = 413.3\,\text{nm}$) are initially offset by $x_0 = y_0 = 50\,\text{mm}$ from the beam center and rotated by $\theta_0 = \phi_0 = -2°$. **a** Snapshots of the beam-riding flexible lightsail's position, angular orientation, temperature, and shape at different times. **b** Lightsail trajectory throughout the 5 s simulation duration. **c** Lightsail $x$- and $y$-position and radial distance $r$ from the beam center versus time, exhibiting bounded oscillation around the equilibrium at $x, y = 0$. **d, e** Evolution of pitch $\theta$ and roll $\phi$, respectively, of the rigid lightsail versus time, showing multi-frequency oscillation around the equilibrium at $\theta, \phi = 0$. **f, g** Distribution of $\theta$ and $\phi$ angles,

respectively, of mesh elements of the flexible lightsail versus time, with the color bar depicting normalized counts on a logarithmic scale with a bin width of 0.1°, showing both bounded oscillations and limited angular spread, with minor shape distortion observed through the range of surface tilt angles at any given time. For **d**–**g**, insets show fast-frequency oscillations within a reduced time window (0.1 s). An animation of this simulation is available as Supplementary Movie 3. In the Supplementary Information, results for passive stabilization of a flexible metagrating lightsail being only initially displaced (Supplementary Fig. 10) but not tilted relative to the beam optical axis are also given, with animation being available as Supplementary Movie 4.

more than 25 times lower than the reported 6.4 GPa tensile limit. Therefore, a meter-sized flexible $Si_3N_4$ lightsail should exhibit mechanical stability in its propulsion phase despite being subject to large thermal gradients, spin tensioning, and nonuniform beam intensity.

The trajectories of flexible and rigid lightsails indicate bounded motion and thus marginally stable dynamics as expected (Fig. 6b). During the simulated propulsion duration, the lightsails remain within 180 cm of the beam center, traversing triangle-like trajectories in the

$x$–$y$ plane. Comparing trajectories of the flexible lightsail and its rigid version, both exhibit similar behavior consistent with marginal stability. Slight deviations in trajectory are revealed more clearly by the oscillatory displacement of the lightsail centers of mass along $x$, $y$, and the radial distance $r$ versus time (Fig. 6c). At first, both flexible and rigid lightsails follow almost indiscernible trajectories. After 0.8 s, differences in $x$ and $y$ become more visible but do not grow continuously over the studied time duration, ruling out the accumulation of numerical error due to insufficiently small time stepping as a possible

reason. Instead, we attribute the small differences in position to the role of shape distortions in flexible lightsails and the effect of thermal expansion.

To elucidate the influence of temperature and thermal strain in the flexible lightsail simulations, we simulated propulsion under conditions of zero absorptivity and emissivity to keep the lightsail temperature constant at 300 K (Supplementary Figs. 14a and 15). The resulting trajectory is again very similar to that of the flexible and the rigid lightsail but does not match either perfectly. However, closer resemblance in dynamics between the thermally inactive flexible lightsail and the rigid lightsail can be observed, suggesting that thermal effects play a bigger role than shape distortions. Both exist due to the nonuniformity of laser illumination, resulting in nonuniform temperature distribution and thermal strain.

Examining the lightsail tilt angles $\theta$ and $\phi$ versus time for the rigid lightsail (Fig. 6d, e), we observe a fast-oscillating component at 240 Hz associated with the 120 Hz spin frequency originating from its two-fold cyclic symmetry (see insets) superimposed upon multiple slower nutation/precession frequencies. Although the pitch and roll angles grow larger than the initial tilt offset, both $\theta$ and $\phi$ remain bounded between ±7°. For the flexible lightsail tilt, we present the distribution of pitch and roll angles for all mesh triangles across the lightsail surface as normalized time-domain histograms in Fig. 6f and g, respectively. The flexible lightsail dynamics are overwhelmingly similar to the rigid-body case and, importantly, are bounded, illustrating the effectiveness of spin stabilization. A closer look at the time-dependent pitch and roll angles reveals that the lightsail does deform (Supplementary Fig. 16), as indicated by an average angular spread of ~1° and a maximum angular spread of almost 14° (Supplementary Fig. 17). This observation emphasizes the importance of simulating flexible lightsails and verifying their self-stabilization in the presence of beam-induced angular deformations given the limited angular range for self-restoring forces and torques to be effective.

While the specific design presented here appears marginally stable for the chosen initial conditions and acceleration duration, substantial deviations from this design and set of assumed parameters can produce unstable behavior. Reducing the spin frequency from 120 to 80 Hz, increasing the beam width from 0.4$D$ to 0.5$D$, or increasing the gap between resonators by 20% for both TE and TM unit cells all result in unstable dynamics (Supplementary Fig. 18), which highlights the importance of carefully choosing the beam width, spin frequency, and optical design for passive stabilization. Specifically, at lower spin frequencies, differences between flexible- and rigid-body dynamics become more apparent, culminating in a situation where a rigid lightsail veers away from the beam while its flexible version remains on a bounded trajectory (Supplementary Fig. 19), emphasizing the importance of modeling mechanical deformations and implying potential benefits for self-stabilizing lightsail acceleration.

We have presented time-domain multiphysics simulations of flexible lightsail membranes undergoing the initial stages of acceleration of up to 5 s due to radiation pressure propulsion. We have explored both the lightsail beam-riding stability and dynamic structural stability by addressing the most relevant physics for flexible lightsail acceleration, including first-order linear elastic behavior, heat flow, and optical scattering. Specifically, we have shown proof-of-concept examples of flexible, meter-scale lightsails, tensioned via spin-stabilization, that exhibit a stable shape without any stiffening elements. While certain concave specularly reflecting lightsail shapes such as paraboloids can enable both beam-riding stability and shape stability, passively stabilized flat lightsail designs based on $Si_3N_4$ metagratings are of particular interest for experimental lightsail development, owing to the mechanical strength, low optical absorption, and ability to be fabricated in planar thin-film form at the wafer scale. Specifically, we have demonstrated that high-speed spin stabilization at 120 Hz is largely effective in rigidifying a flexible

metagrating-based lightsail to exhibit quasi-rigid dynamics, with subtle differences appearing due to structural deformations and thermal effects. Importantly, beam-induced shape deformations on the order of degrees and millimeters perturb but do not disrupt the beam-riding dynamics of such flexible, meter-sized, and subwavelength-thick lightsails enabled by optical metastructures.

Further optimization of the metagratings and lightsail structure, potentially by including other materials, will be necessary to meet the nominal design targets proposed by the Breakthrough Starshot program and to produce a full-scale lightsail design for interstellar missions. Our design represents an important first step towards this goal, and the simulation tools reported here will likely be useful in achieving this goal. Future work should be directed towards modeling, implementing, and experimentally probing the temperature dependence of optical reflectivity, absorptivity, and emissivity, to better understand the upper limits of achievable acceleration−a key factor in determining the viability of interstellar exploration via laser-propelled lightsails. Other second-order effects may also be worthy of investigation, such as the effects of strain on optical properties. Additionally, our simulation approach may be useful in addressing other challenges for interstellar lightsail development, such as payload integration and codesign of the propulsive laser system.

As it is difficult to infer absolute stability from time-domain simulations of marginally stable lightsails, a more useful future application of our approach might be lightsail improvement and optimization. Our present lightsail patterning was selected based on parametric optimization under rigid-body Floquet theory, but the complexity of flexible lightsail dynamics suggests that a more advanced optimization approach based on numerical time-domain simulations may yield more favorable designs with increasingly complex building blocks and physical behaviors being modeled. Nevertheless, studying the initial seconds of lightsail acceleration provides valuable insight into flexible lightsail design. Therefore, we share our simulation code[64] to further expand efforts by the lightsail community to develop new and improved designs for interstellar propulsion, optical levitation, and long-range optical manipulation of macroscopic objects.

## Methods
### Flexible lightsail modeling
A mass-spring mesh model is constructed for the linear elastic behavior of a lightsail membrane, and a finite-difference time-domain approach is employed to simulate its dynamics. Within the mesh, each vertex is assigned a mass based on the local membrane thickness, the area of the adjoining triangles, and the material density. Elastic behavior of the membrane is captured by the edges, each of which is assigned a linear elastic coefficient (spring constant) based on the mesh geometry and local material properties. Due to the extreme aspect ratios of the lightsail membranes, having meter-scale extent and submicron thickness, bending stress is negligible compared to the in-plane stresses induced by the chosen spin frequencies, and bending stiffness is negligible compared to the out-of-plane forces induced by optical propulsion and membrane elastic deformation. Therefore, bending stiffness is omitted from the model, and the specific shear modulus of the material is also neglected. This approach provides reasonable first-order insights into the behavior of ultrathin membranes under tensile loading, which is the predominant type of loading in lightsail applications.

Mesh vertices and edges are also used to compute temperature and thermal conduction, respectively, and linear thermal expansion is effected by considering local temperatures when calculating edge strain. The mesh triangles are used to compute optical forces and absorption, as well as radiative cooling and radiative heat transfer, all of which are distributed to the adjoining mesh vertices to enable the resulting changes in momentum and temperature to be determined.

## Burst diameter and maximum spin frequency

The stationary burst diameter $D_{SB}$ introduced in Table 1 and discussed in the main text for a clamped circular membrane is calculated according to the following formula[54]:

$$\sigma_{max,clamped} = 0.423 \sqrt[3]{\frac{E(\Delta p)^2 R^2}{t^2}} \rightarrow D_{SB} = 2R\left(\sigma_{max,clamped} = \sigma_T\right)$$
$$= \frac{2t}{\Delta p \sqrt{E}} \left(\frac{\sigma_T}{0.423}\right)^{3/2}, \quad (1)$$

where $\sigma_T$ is the tensile strength of the material under consideration, $t$ the membrane thickness corresponding to an areal density of $0.1\,g\,m^{-2}$ and $\Delta p = 67$ Pa the pressure differential between both sides corresponding to an effective photon pressure of $10\,GW\,m^{-2}$ illumination for unity reflectance.

The second figure of merit, the maximum spin frequency $f_{max}$ of a circular disk, is calculated according to[55]

$$\sigma_{max,spinning} = \frac{3+\nu}{8} \rho \omega^2 R^2 \rightarrow f_{max} = f\left(\sigma_{max,spinning} = \sigma_T\right)$$
$$= \frac{1}{2\pi R} \sqrt{\frac{8}{3+\nu} \frac{\sigma_T}{\rho}}, \quad (2)$$

with the disk radius $R$ corresponding to a lightsail area of $10\,m^2$, $\nu$ being the material's Poisson ratio, and $\rho$ being the material density.

## Metagrating-based lightsail design

As shown in Fig. 5a, a circular lightsail is partitioned into four sectors, forming two orthogonal pairs of symmetrically opposed wedges. We assume a linearly polarized incident beam, with its electric field aligned with the body-frame y-axis $y_{BF}$. Thus the blue sectors (1/3 of the lightsail area) experience transverse-electric (TE) polarization, and the brown sectors (2/3 of the lightsail area) experience transverse-magnetic (TM) polarization, and the specific asymmetric metagratings for each sector provide stabilizing forces and torques for their respective design plane and polarization. For spin-stabilized lightsails, we assume that the beam polarization rotates synchronously with the spinning lightsail. Electromagnetic simulations were performed to determine the optical response of the TE and TM metagrating unit cells.

## Electromagnetic simulations

Electromagnetic response of the TE and TM metagrating designs for a laser propulsion wavelength of $\lambda = 1064$ nm were simulated in COMSOL Multiphysics assuming periodic Floquet boundary conditions. For high-stress stoichiometric silicon nitride[45], we assume a refractive index of $Re(n) = 2$ and an extinction coefficient of $Im(n) = 2 \times 10^{-6}$ at $\lambda = 1064$ nm. The TE and TM metagrating unit cells shown in Fig. 5a are defined by $w_1^{TE} = 600$ nm, $w_1^{TM} = 520$ nm, $w_2^{TE/TM} = 200$ nm, $d^{TE} = 1600$ nm, $d^{TM} = 1350$ nm, $g^{TE} = 190$ nm and $g^{TM} = 200$ nm. The resonators' height and substrate thickness are chosen to be $h = 400$ nm and $t = 200$ nm, respectively. The process of identifying these self-stabilizing unit cell designs was based on linearized stability analysis. While nonspinning designs are marginally stable if the eigenvalues of the Jacobian matrix derived from the lightsail equations of motion are purely imaginary, for spinning lightsails as linear-time periodic systems, we must employ Floquet theory to assess the stability of the designs[61,62]. Specifically, our chosen unit cell designs for lightsails spinning at 120 Hz produce absolute values of eigenvalues of the monodromy (state transition) matrix equal to 1, i.e., $|\lambda_i| = 1$, which is a sufficient and necessary condition for marginal stability. More information about this analysis can be found in the Supplementary Information (see Supplementary Note 6).

Except for the resonator height and substrate thickness, all geometrical parameters were varied systematically to select and compare suitable metagrating designs. By sweeping the incidence angle between $\pm 25°$ for both pitch ($\theta$) and roll ($\phi$) tilt, angle-dependent optical pressures can be obtained via integration of the Maxwell Stress tensor around the respective unit cell.

## Time-domain simulations

We used the exported look-up tables of optical pressures as inputs to our rigid and flexible membrane dynamics simulations. In the former case, (normalized) optically induced forces and torques over a range of incidence angles ($\theta$, $\phi$) and translational offsets ($x$, $y$) can be derived assuming a Gaussian beam characterized by its peak intensity $I_0$ and beam width $w$. For the flat metagrating-based lightsails, a Gaussian beam with a peak intensity of $I_0 = 1\,GW\,m^{-2}$ and a width equal to 40% of the lightsail diameter, i.e., $w = 0.4D = 0.4$ m, was assumed. In the case of rigid lightsails, for a given set of initial conditions (position, velocity, angular orientation, and angular frequency), the coupled equations of motion were evolved numerically using MATLAB's ode45 solver (relative tolerance of $10^{-6}$) to obtain the trajectory and time-dependent displacement and tilt of propelled rigid lightsails described by their centers of mass. Normalized relevant quantities can be converted to real-life values by specifying $I_0$, the lightsail diameter $D$, and calculating the normalized time constant $t_0 = \sqrt{mc/I_0}$, where $m$ is the total mass of the lightsail.

To simulate the acceleration of flexible lightsails, and to assess their stability, we implement a finite-difference time-domain approach wherein the forces and heat flow are calculated at each mesh vertex, and the resulting changes in position, velocity, and temperature are calculated explicitly over a time step $\Delta t$. We select the mesh density to be sufficiently fine and $\Delta t$ to be sufficiently small so as to capture the membrane dynamics and vibrational modes with reasonable fidelity and perform control simulations to ensure that changes to time stepping and mesh discretization do not substantially change the results. The dynamics of curved lightsails were simulated using symplectic Euler integration (Figs. 3 and 4), whereas the flat flexible lightsails were simulated via numerical integration based on the Runge-Kutta method (Figs. 5 and 6, see also Supplementary Note 3).

Perceptually uniform, undistorted color maps were used for Figs. 5, 6, Supplementary Figs. 10, 11, 16, 18, and Supplementary Movies 3, 4[65,66].

## Data availability

The relevant data of this study are included in the paper and Supplementary Information file, and raw data are available from the corresponding author upon request.

## Code availability

The MATLAB code for this study has been made available on GitHub at https://github.com/Starshot-Lightsail.

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

## Acknowledgements

The authors acknowledge helpful discussions with James Benford, Victor Brar, Artur Davoyan, Ognjen Ilic, Phillip Jahelka, Adrien Merkt, Lior Michaeli, John Sader, Cora Wyent, and Joeson Wong. We thank Zachary Manchester for insights on numerical techniques and guidance with the Floquet stability analysis. Funding was provided by the Air Force Office of Scientific Research under grant FA2386-18-1-4095 (R.G., M.D.K., and H.A.A.) and the Breakthrough Starshot Initiative (R.G., M.D.K., and H.A.A.).

## Author contributions

R.G. performed the electromagnetic simulations of the metagrating designs and studied the self-stabilizing properties of flat lightsails. M.D.K. developed the flexible mesh simulator, investigated materials and structural stability, and studied the curved lightsail shapes. H.A.A. supervised the investigation. All authors participated in drafting and reviewing the manuscript.

## Competing interests

The authors declare no competing interests.
