## [Peer Review File · Nature Communications]

Dynamically Stable Radiation Pressure Propulsion of Flexible Lightsails for Interstellar ExplorationREVIEWER COMMENTS

Reviewer #1 (Remarks to the Author):

The authors present an interesting study of the stability of spinning and stationary membranes under acceleration by an intense laser beam. The paper advances the field by considering the complex deformations of thin membranes and demonstrating the effect of spinning at rates close to the theoretical maximum. Before I can recommend the paper, I would like the authors to address the following concerns and suggestions:

- 1) On page 6, the authors mention “collapse by elongation”. It would be helpful to illustrate (perhaps in Fig. 1) and explain the mechanism in more detail for the broader audience.
- 2) On page 7, the authors write “The assumed ‘stationary’ perimeter constraint overestimates required membrane tensile strength, since any viable perimeter structure must have an extremely small mass accelerated along with the lightsail.” I would suggest rephrasing or expanding this sentence since the meaning is not clear.
- 3) Also on page 7 and in Table 1, the authors list “maximum spin speed f_{max} ”. First, I would suggest changing from “speed” to “frequency”. Also please specify what formulas were used to derive this max spin speed/frequency.
- 4) On page 9, the authors write “the opposite side of the lightsail counteracts the restoring forces”. Could the authors explain this point more and perhaps illustrate it with a diagram?
- 5) On page 11, the authors mention “nematic liquid crystals”. Are these materials appropriate for the lightsail applications given the extreme light intensities and high temperatures?
- 6) On page 13, the authors write “no visibly apparent shape distortion thanks to the sufficiently large tensioning forces arising from spin-stabilization”. How large are the actual displacements from the flat shape (e.g., in the center)? Also, what are the largest angular deviations from a flat shape? On page 14, they suggest the average angular deviation is on the order of 1 degree but it would be great to know the maximum value as well.
- 7) The assumed intensity of the incident laser beam varies from 10 to 4 to 1 GW/m^2 in various parts of the paper. What is the rationale for assuming such different values of laser intensity in different sections?
- 8) On page 14, the authors report the maximum tensile stress of ~ 246 MPa for a Gaussian beam with a peak intensity of $I_0 = 1 \text{ GW m}^{-2}$ and a width equal to 40% of the lightsail diameter. These results suggest that the intensity can be increased further without the sail failing. Could the authors report at what intensity the sail fails and the actual failure mechanism?
- 9) Can the authors comment on the acceleration distance to $0.2c$ for these cases and how it compares to other literature reports?
- 10) Finally, in the captions of Figs. 3-6, could the authors specify the material and the thickness of the membrane simulated for each figure?

Reviewer #2 (Remarks to the Author):

This work tackles the problem of a sail structure driven by radiation pressure of laser light. The goal is for a very thin (sub-micron) and large area (meter-scale) sail to reach relativistic velocities under illumination of powerful lasers, taking into account the flight of the sail, structural stability, and heating effects. The work develops a mesh-based simulator tool to model the dynamical behaviour of sails of different shapes, such as curved sails and planar sails. It is found that spinning curved light sails can maintain their shape during acceleration, and that non-curved (planar) structures can also be stable when silicon nitride elements are integrated to control the reflection of light. The simulator tool produces very nice visuals of sail shape evolution and deformation during acceleration. The code is made freely available which is highly commended. The manuscript does an excellent job to describe the background of the problem and the associated challenges of relativistic laser propulsion flight which include material tensile and temperature limits. The paper is well-written, clear, and easy to read.

The main concern is the originality and the significance of this work. Analysis of light sail structures for stability is not new: for curved sails (e.g. [21,22,23]) and for non-curved planar sails with nano patterned surfaces (e.g. [18,25,27,28] and by the authors [24,33]). The insight on flying stability provided by this work appears to be limited, since the same materials and meta grating pattern have been previously reported for stability behaviour. The simulator tool for evaluating the structure stability of the sail is novel, but the significance of this tool is not quite clear. For a spinning sail, the points of maximum stress are intuitive (i.e., the centre of the flat sail) and can be calculated and compared to material tensile limits without the tool. Looking at the planar silicon nitride sail and applying textbook equations for stresses near the centre of a spinning disk (radial stress = tangential stress = $(3+\nu)/8 \cdot \rho \cdot \omega^2 \cdot r^2$) gives total stress of ~ 222 MPa, very similar to the value from the simulator that confirms the finding that the stress is many times lower than tensile limit. The radiation pressure stresses and structural spinning stresses are quite different in magnitude in laser sails, so while the simulator can solve for both, it isn't clear that that simulator feature offers significant new insight particularly for planar sails. As the authors state, a flexible planar sail (which needs the simulator) has very similar dynamics as a rigid sail which can be modelled without the mesh-based simulator tool. It seems the tool can be useful to identify failure/crumpling during acceleration for curved sails, but that seems to be of comparatively moderate importance. Given these observations on novelty and significance, I am not able to recommend publication in Nature Communications.

Additional comments:

It is stated that polyimide is far too weak to span meter-scale areas between structural supports, but it is not quite clear why that would be the case. The stationary burst diameter refers to the membrane rigidly clamped at its perimeter, where the perimeter is also clamped and stationary. But the perimeter of a light sail moves with the sail, so in principle a pressure of 67 Pa could be supported for large polyimide sails, especially if uniformly applied. Authors do note that the perimeter constraint overestimates the required tensile strength, but I am not sure the suggestions provided by this metric are useful for screening materials. One possible alternative is to discuss maximum tensile stresses generated under non-uniform/gaussian pressure loads on freely moving sails.

Curved sails in Figs. 3 & 4 have the added complexity that there are multiple reflections for radiation thrust and also thermal radiation (the inner surface radiation configuration factor is not zero). The latter would lead to additional heating. It is not clear if the temperature calculations account for that. If not, there is possibly a simple way to estimate the temperature discrepancy (to first order, relative scaling with a configuration factor could account for the fact that parts of the sail emit/receive thermal radiation into/from other parts of the sail and not just space)

Figure 3-top-right: it can be observed that a paraboloid sail with spin 90 Hz remains bounded for longer than flat sail (blue, red) or paraboloid sail (orange, non-spinning). Does this observation depend on the chosen starting conditions? It would be helpful to discuss if/how starting conditions affect these observations.

The last sentence in the abstract that refers to the lightsails and their fabricability (“...and that their [membrane-like lightsails] fabrication should be within the reach of modern microfabrication technology”) is not quite correct. The work shows nano-element features that have dimensions that can be fabricated. But that does not support the claim that light sails of needed meter-scale and sub-micron thickness are, at present, within the reach of fabrication technology. Clarifying the difference between the two would be helpful.

In Fig. 6f,g, the scalebar uses “Norm(Log(Max(Counts,10/10)))” units. It would be helpful to describe the scale bar units either in the caption or in the main text closer to where the figure is referenced.

Reviewer #3 (Remarks to the Author):

In this work, the authors present a detailed calculation of the motion of laser-propelled lightsails, and demonstrate many sail configurations that exhibit stable motion, where the sail remains within the drive laser beam, and does not crumple or decompose. This is a very complete study, which considers the thermal and mechanical properties of the constituent materials, multiple sail geometries, and - for the first time that I am aware of - the potential bending motion and internal tension of the sail under acceleration. The consideration of these latter two parameters make this work exemplary and very interesting to a broad community of scientists who either work directly with laser sails, or have looked on with interest wondering if they are at all feasible. By explicitly including bending and potential fissuring of a flexible sail, the authors have shown unequivocally that stable laser sails are possible, even without a rigid structure, and for very thin membranes. As such, I fully endorse this work for publication in Nature Communications.

I do, however, feel this manuscript could be improved by a few inclusions and explanations:

1 - The authors explicitly state that bending stiffness should be 'negligible', and is not considered in their 2D mesh geometry, but it is not immediately obvious to me that this is the case. I would naively expect that bending stress would be a bigger problem than in-plane tension. Could the author's include a rough estimate of what the bending stress is for the flexible, spinning SiN membrane, described in Fig. 6 (the seemingly most promising design)? I understand that full scale simulations are beyond the scope, but it would be insightful if they could make a rough calculation that

considers the sail curvature and local forces at a particular point in time.

2 - In Ref. 21, spinning conical sails are considered and the author of that work explicitly states that if the axis of angular momentum is not aligned with the laser beam, the sail would be unstable. That work also provides a brief explanation describing why that is the case. In this work, the authors do explicitly mention the problem described in Ref. 21, but it is unclear how it is resolved in this work - i.e. why the spinning cases seem to avoid destabilization. Can the authors provide some explanation?

3 - The precise method for modeling the motion of the sail is not completely clear. The authors state they model the dynamics in timesteps that are 10 - 20x shorter than the smallest periodic motion, but otherwise it is not clear what exactly is being done. Are they using a Runge-Kutta method?

4 - Many papers in this field parameterize sail performance in terms of acceleration distance. Could the authors calculate the acceleration distance for their sails shown in Fig. 6 and include those numbers in the manuscript?

SUMMARY OF REVISIONS AND AUTHORS' RESPONSE TO REVIEWERS

First, we would like to thank the reviewers for their careful assessment of our work and for their constructive and insightful comments. In this response letter, we address every comment and concern of the referees and highlight the resulting changes to the manuscript and supplementary information accordingly. We summarize high-level changes on the first three pages, then provide detailed point-by-point responses to each reviewer comment, formatting our responses in blue for ease of reading. We also quote specific passages from the manuscript that were changed or added in response to the reviewers' comments, highlighted in yellow. The modified manuscript has been resubmitted as a marked-up version showing all changes.

Summary of Revisions to the Manuscript:

- a) We replaced Fig. 3 with an updated version based on simulation results obtained from the most recent version of our code. Changes pertain to specific simulation parameters (i.e., more accurate emissivity and mesh generation). Importantly, none of our conclusions are affected by this replacement.
- b) We modified Fig. 4 to additionally illustrate the effect of counteracting forces from multiple reflections in curved lightsails following comment #4 by reviewer 1.
- c) We replaced Fig. 6 with new simulation results obtained with the Runge Kutta method. We note that the new results do not change our conclusions on the self-stabilization of flexible metagrating-based lightsails and their comparison to rigid-body dynamics.
- d) Additionally, we revised Fig. 6f and Fig. 6g to highlight the flexible nature of the spinning lightsail by showing the range of angular deformations. We also added a short description to the caption as suggested by reviewer 2.
- e) We revised the main text to emphasize the importance and significance of modelling spinning (flat) lightsails as flexible membranes in response to reviewer 2.
- f) We rephrased and clarified the motivation for the figure of merit “stationary burst diameter” to address the comments raised by reviewer 1 and reviewer 2.
- g) We included a brief comment on the dependence of variously shaped curved lightsails presented in Fig. 3 on the chosen starting conditions as suggested by reviewer 2.
- h) To address reviewer 1's comment on bending stiffness, we added a brief explanation for its omission in our simulations based upon calculations and references to spinning thin elastic disks and membranes.
- i) As suggested by reviewer 3, we now state the numerical integration method used for our simulations in the Methods section.
- j) We revised the final sentence of our abstract to clarify the fabricability of membrane-like lightsails in response to reviewer 2.

- k) We added the formulas for the stationary burst diameter and maximum spin frequency to the Methods section following the suggestion by reviewer 1.
- l) We now specify the material and thickness of simulated lightsails in Fig. 3 – 6 as recommended by reviewer 1.

Summary of Revisions to Supplementary Information:

- a) Due to main text changes **a) – d)**, we regenerated Supplementary Videos 1 – 4. Importantly, Supplementary Videos 3 and 4 now show lightsail dynamics color-coded with displacement from a flat shape, supporting our emphasis on the importance of shape modelling of flat flexible lightsails.
- b) We added a new note on the stationary burst diameter figure of merit (Supplementary Note 2), in which we clarify its motivation, but also compare its burst intensities for perimeter-supported, free-flying lightsails under Gaussian illumination conditions using our simulator.
- c) We included a new section to Supplementary Note 3, elaborating on our implementation of ray tracing for curved specular lightsails, accompanied by ray-tracing plots.
- d) We added another section to Supplementary Note 3, in which we provide details on the numerical integration method used for simulation of flexible lightsails in response to comment #3 by reviewer 3.
- e) We wrote a new note (Supplementary Note 4) with two additional figures to describe the implementation of a radiative heat transfer model in our simulator for parabolic lightsails with secondary reflections to address the comment raised by Referee 2.
- f) In the new Supplementary Note 5, we also describe the simulation results of a paraboloid Si_3N_4 lightsail that is *not* destabilized by secondary reflections.
- g) We added a new note (Supplementary Note 9) to explain the calculation of and illustrate the displacement from a flat shape for flexible lightsails in response to comment #6 by reviewer 1.
- h) In response to the same comment #6 by reviewer 1, we also provide calculations and visualization of the angular deviation of the flexible metagrating-based lightsail from a flat shape.
- i) To aid our argument of multiphysics modelling and specifically the role of thermal strain, we revised Supplementary Fig. 13 to include the strain of a thermally inactive flexible lightsail with the optical design reported in Fig. 5 and otherwise identical initial conditions as those noted for Fig. 6.

- j) We added Supplementary Note 14 with a short discussion of the acceleration performance and distance of our studied lightsail designs as suggested by reviewer 1 and reviewer 3.

Summary of other changes:

- a) In the previously submitted version of the manuscript, we regrettably swapped Fig. 6 with Supplementary Fig. 3. These are similar simulations – one assumes the lightsail being misaligned in position only, the other assumes misalignment of the lightsail in *both* position and tilt. The figures now appear in the correct location to accompany their captions. We apologize for confusing the figures in our prior submission.
- b) We have unified our previously differing definitions of beam size reported throughout the manuscript.
- c) We reran many simulations (and recreated accompanying figures) to ensure greater consistency between stated and unstated parameter values. In particular, we re-evaluated our selection of optical and thermal properties for Si membranes, based on recent publications discussing thermal runaway in Si lightsails. The manuscript and supporting information now report simulation parameters more clearly and consistently, and discuss in greater detail the rationale for selection of models and materials properties.
- d) The simulation code has been continuously improved to support more comprehensive physics and improved computation methods; the updated figures reflect results obtained with the latest version of the code library. An important benefit of this effort is that we have now included the exact simulation codes used to generate each figure as examples in the publicly released code on GitHub, ensuring that all parameter values, computation methods, and assumptions are directly available to readers of the manuscript.

RESPONSE TO REVIEWER COMMENTS

Reviewer #1 (Remarks to the Author):

The authors present an interesting study of the stability of spinning and stationary membranes under acceleration by an intense laser beam. The paper advances the field by considering the complex deformations of thin membranes and demonstrating the effect of spinning at rates close to the theoretical maximum. Before I can recommend the paper, I would like the authors to address the following concerns and suggestions:

We thank the referee for the encouraging evaluation of our work and the insightful comments, which have improved our manuscript.

1) On page 6, the authors mention “collapse by elongation”. It would be helpful to illustrate (perhaps in Fig. 1) and explain the mechanism in more detail for the broader audience.

We have expanded the description of “collapse by elongation” in the main text by providing a brief explanation. Moreover, we now refer to Fig. 1b specifically in this sentence, which illustrates this effect.

However, open concave shapes such as cones and paraboloids are still prone to collapsing by elongation (Fig. 1b, center left). When curved lightsails become slightly deformed, their elongated regions present larger cross-sectional area and smaller incidence angles, resulting in an increased effective photon pressure, whereas the narrowing regions similarly experience decreasing effective photon pressure, furthering the distortion and leading to collapse.

2) On page 7, the authors write “The assumed ‘stationary’ perimeter constraint overestimates required membrane tensile strength, since any viable perimeter structure must have an extremely small mass accelerated along with the lightsail.” I would suggest rephrasing or expanding this sentence since the meaning is not clear.

We have improved this section to clarify this issue, also in response to a related comment made by reviewer #2.

3) Also on page 7 and in Table 1, the authors list “maximum spin speed f_{\max} ”. First, I would suggest changing from “speed” to “frequency”. Also please specify what formulas were used to derive this max spin speed/frequency.

We thank the reviewer for noting the inadvertent omission of the formulas and for the suggested improvement to our terminology, which we have adopted throughout the text. In addition, we included the formula for the maximum spin frequency in the Methods section.

The second figure of merit, the maximum spin frequency f_{\max} of a circular disk, was calculated according to⁵⁰

$$\sigma_{\max} = \frac{3 + \nu}{8} \rho \omega^2 R^2 \rightarrow f_{\max} = f(\sigma_{\max} = \sigma_T) = \frac{1}{2\pi R} \sqrt{\frac{8}{3 + \nu} \frac{\sigma_T}{\rho}}$$

With the disk radius R corresponding to a lightsail area of 10 m^2 and ρ being the material density.

4) On page 9, the authors write “the opposite side of the lightsail counteracts the restoring forces”. Could the authors explain this point more and perhaps illustrate it with a diagram?

We have revised this sentence to clarify the explanation. This effect was also illustrated in Supplementary Video 2, to which we now also directly refer in the main text. Moreover, we added the same illustration to Fig. 4 as Fig. 4a.

Secondary reflections do increase the total photon pressure on the lightsail, resulting in faster acceleration, but reflected light striking the opposite side of the lightsail induces additional forces and torques there, which counteract the restoring forces and torques produced by the first reflection, thus destabilizing the lightsail (Fig. 4a).

Copied from Supplementary Video 2 to illustrate the potential destabilization effect of multiple internal reflections in curved lightsails.

5) On page 11, the authors mention “nematic liquid crystals”. Are these materials appropriate for the lightsail applications given the extreme light intensities and high temperatures?

In describing nematic liquid crystals, we refer to the exciting advances previously reported that experimentally validate the trajectory-restoring behavior due to optical diffraction from liquid crystal-based gratings. We acknowledge the maturity, large-scale manufacturability, and potential for electro-optical modulation of liquid crystal films in solar sail applications, used for example in the IKAROS spacecraft. We also believe that laser-driven lightsail applications for relativistic space travel will require materials that are thermally and mechanically more robust. The reviewer is correct in stating that extreme light intensities of up to 10 GW/m^2 will be needed for acceleration to relativistic velocities within the targeted duration and distance, which demands for absorption of less than 10^{-6} and high tensile strength. Silicon nitride has a path to meeting these extreme requirements which currently appears to be shorter than that for nematic liquid crystals.

6) On page 13, the authors write "no visibly apparent shape distortion thanks to the sufficiently large tensioning forces arising from spin-stabilization". How large are the actual displacements from the flat shape (e.g., in the center)? Also, what are the largest angular deviations from a flat shape? On page 14, they suggest the average angular deviation is on the order of 1 degree but it would be great to know the maximum value as well.

We thank the reviewer for the suggestion. In Fig. 6f and 6g, we plot the distribution of pitch angle θ_m and roll angle ϕ_m of every mesh element m at every time step t_i according to Euler angles (ψ, θ, ϕ) based on the 1-2-3 or x - y '- z ' convention of rotations. To obtain the angular deviations from a flat shape, we therefore need to subtract θ_m and ϕ_m from the averaged pitch and roll corresponding to a flat shape, i.e., $\bar{\theta} = \sum_m \theta_m / m$ and $\bar{\phi} = \sum_m \phi_m / m$. In doing so, we obtain a maximum pitch deviation $|\Delta\theta| = \max|\theta_m - \bar{\theta}| \approx 7.8^\circ$ and maximum roll deviation $|\Delta\phi| = \max|\phi_m - \bar{\phi}| \approx 8.5^\circ$ from the corresponding flat shape at that time step as can be seen from the figure below, which is now part of a new Supplementary Fig. 16.

As for the actual displacements from the flat shape, we again define the flat shape at time t_i according to the averaged pitch, roll and yaw angles $\bar{\theta}, \bar{\phi}, \bar{\psi}$, respectively, and the center-of-mass coordinates $\mathbf{r}_{\text{COM}} = (x_{\text{COM}}, y_{\text{COM}}, z_{\text{COM}})$. With this, we can then calculate the actual displacements of each node in our mesh (or alternatively centroid defined by three nodes) from that flat shape (see newly added Supplementary Note 9). Below, we show the maximum displacement from a flat shape versus time, from which we find it to be 2.96 mm (node displacement) or 2.7 mm (centroid displacement). This figure has also been included as a new Supplementary Fig. 11.

Finally, the location of maximum displacement from a flat shape depends on the current lightsail-beam alignment both in terms of translation and rotation as well as the induced membrane mechanics, e.g., deformations and modes. In general, we observe beam-induced shape deformations with positive displacements (away from the laser source) near the center of the lightsail and negative displacements at the edge of the lightsail, as one would intuitively expect during the acceleration phase. This can be seen from color-coded surface plots of displacement of nodes from a rigid lightsail (flat shape) within the flexible lightsail reported in Fig. 6 at various time steps, which we included as new Supplementary Fig. 10.

7) The assumed intensity of the incident laser beam varies from 10 to 4 to 1 GW/m^2 in various parts of the paper. What is the rationale for assuming such different values of laser intensity in different sections?

The maximum intensity of $10 \text{ GW}/\text{m}^2$, which was assumed for calculation of the stationary burst radius, represents the targeted laser intensity for the Breakthrough Starshot mission as reported previously by our group (Atwater et al., Nature Materials, 2018).

For simulated lightsails in the present manuscript, we selected optical designs and acceleration conditions that would not lead to implausibly high temperatures. We note that our

understanding of the optical properties and thermal limits of the constituent materials has evolved during the project. More recently, Holdman et al. (Advanced Optical Materials, 2022) showed that silicon lightsails risk thermal runaway at temperatures exceeding ~ 500 K, and thus likely cannot endure laser intensities exceeding ~ 5 GW/m² due to two-photon absorption (even though their calculations considered a slightly different optical structure). Seeking to present a better rationale for beam intensity choice during manuscript revision, we performed new simulations for Fig. 3 and Fig. 4 using the recent optical data from the aforementioned publication by Holdman et al. Assuming that photonic engineering efforts could improve the emissivity of the membrane to 0.8 (but not proposing or modelling a specific structure), we conclude that the Si paraboloids could potentially operate at up to the two-photon absorption limit of 5 GW/m² and use this value for our illustrative examples. Importantly, these thermal and intensity thresholds are exceeded once multiple reflections are considered, which illustrates the importance of developing rigorous multiphysics models to simulate all pertinent behavior).

Finally, for the flat silicon nitride lightsail reported in Fig. 5 and Fig. 6, assuming an extinction coefficient of 2×10^{-6} as reported by Karuza et al. (Physical Review A, 2013), absorption values of 9.45×10^{-6} and 9.76×10^{-6} were calculated for our Si₃N₄ TE and TM metagratings, respectively. Their larger absorption required reducing the incident laser beam intensity further for thermally viable simulations of initial acceleration dynamics, hence the third choice of 1 GW/m², causing the temperature of our silicon nitride lightsail to rise by ~ 700 K.

We note that none of our reported lightsails are meant to represent finalized, Starshot-compatible designs, as we focused on characterizing their stabilized or destabilized dynamics within the first few seconds of propulsion. Specifically, absorption well below 10^{-5} could be achieved for the LPCVD Si₃N₄ metagratings with improvement to material deposition and processing, for example, via repeated thermal annealing to further reduce hydrogen impurities-related absorption losses (see Liu et al., Nature Communications, 2021).

8) On page 14, the authors report the maximum tensile stress of ~ 246 MPa for a Gaussian beam with a peak intensity of $I_0 = 1$ GW m⁻² and a width equal to 40% of the lightsail diameter. These results suggest that the intensity can be increased further without the sail failing. Could the authors report at what intensity the sail fails and the actual failure mechanism?

The reviewer correctly observed that the peak laser intensity could be increased beyond the assumed 1 GW/m² for the flat spinning lightsail. The main failure mechanism at increased laser intensities would be thermal failure, wherein the temperature of the lightsail due to heat absorption exceeds the temperature for vacuum decomposition. In the specific case of the flat spinning Si₃N₄ lightsail referred to by the reviewer, given the assumed extinction coefficient of $k = 2 \times 10^{-6}$, the lightsail would reach the vacuum decomposition temperature of 1673 K after less than 25 ms of acceleration by a laser beam of 10 GW/m² peak intensity. If the peak laser intensity is lowered to 8.5 GW/m², our simulations will predict a maximum temperature of ~ 1600 K, i.e., close to the calculated temperature for decomposition of Si₃N₄. However, we note that the self-stabilizing behavior of the composite metagratings for a lightsail spinning at a specific spin frequency depends on the peak laser intensity due to time being normalized by $I_0^{-1/2}$. While the presented optical design was found using Floquet theory under the assumption of 1 GW/m² peak intensity, other laser intensities might result in dynamical instability.

Finally, we note that mechanical failure appears to be the least concern due to the high tensile strength of the Si_3N_4 lightsail membrane and the fact that for spinning membranes of such sizes, bending stiffness is negligible and the in-plane stress distribution is dominated by centrifugal forces, as we clarify further in response to comment #1 from reviewer #3. Specifically, even at 10 GW/m^2 , which is the target laser power density for the Starshot mission, the simulated strain is less than 10% of the maximum strain limit imposed by silicon nitride's tensile strength and modulus.

9) Can the authors comment on the acceleration distance to $0.2c$ for these cases and how it compares to other literature reports?

The acceleration distances to $0.2c$ of all reported cases in our manuscript will fall short of the targeted acceleration distance by the Breakthrough Starshot 'point design' mission. Our reported silicon and silicon nitride lightsails do not represent finalized designs, as we have not attempted to optimize their optical and structural properties for maximized reflectance over the Doppler-broadened propulsion band and minimized mass, respectively. In comparison with other literature reports such as 1.9 Gm for an inversely optimized 1D grating-based lightsail (Jin et al., ACS Photonics, 2020), ~ 13.5 Gm for a diffractive beam-riding lightsail based on multi-objective optimization (Salary and Mosallaei, Advanced Theory and Simulations, 2021), 23.3 Gm for a thermally viable lightsail (Brewer et al., Nano Letters, 2022), and 23.5 Gm for a similar self-stabilized lightsail including additional optical communication functionality by Taghavi et al. (Nanoscale Advances, 2022), the acceleration distance of our curved silicon and flat silicon nitride lightsails will be one order of magnitude (~ 100 's Gm) and two orders of magnitude (~ 1 Tm) higher. However, we emphasize that achieving lightsail designs with viable acceleration distances comparable to other reports was not the focus of this work.

Instead, our goal for this manuscript has been to address the open question of how structural flexibility necessitated by the ultrathin nature of relativistic lightsails could affect their initial acceleration dynamics by introducing an open-source time-domain simulator for studying the beam-riding behavior of flexible, spinning lightsails, during which we can identify apparent mechanical, thermal or dynamical failure mechanism. Specifically, we have presented exemplary cases of curved and flat lightsails that either become destabilized in their motion due to insufficient spinning or unsuitable optical scattering or achieve beam-riding behavior via their shape or integrated photonic designs. We anticipate that the reported techniques can be employed and combined with further future design optimization to produce lightsail designs that are viable for key specific applications, e.g., the Starshot $0.2c$ 'point design' mission.

10) Finally, in the captions of Figs. 3-6, could the authors specify the material and the thickness of the membrane simulated for each figure?

We thank the reviewer for this suggestion, we have specified the material and included the membrane thickness for simulated results shown in Fig. 3 to 6.

Reviewer #2 (Remarks to the Author):

This work tackles the problem of a sail structure driven by radiation pressure of laser light. The goal is for a very thin (sub-micron) and large area (meter-scale) sail to reach relativistic velocities under illumination of powerful lasers, taking into account the flight of the sail, structural stability, and heating effects. The work develops a mesh-based simulator tool to model the dynamical behaviour of sails of different shapes, such as curved sails and planar sails. It is found that spinning curved light sails can maintain their shape during acceleration, and that non-curved (planar) structures can also be stable when silicon nitride elements are integrated to control the reflection of light. The simulator tool produces very nice visuals of sail shape evolution and deformation during acceleration. The code is made freely available which is highly commended. The manuscript does an excellent job to describe the background of the problem and the associated challenges of relativistic laser propulsion flight which include material tensile and temperature limits. The paper is well-written, clear, and easy to read.

We thank the reviewer for the supportive evaluation of our simulator and manuscript and for the insightful comments on our presented results.

The main concern is the originality and the significance of this work. Analysis of light sail structures for stability is not new: for curved sails (e.g. [21,22,23]) and for non-curved planar sails with nano patterned surfaces (e.g. [18,25,27,28] and by the authors [24,33]). The insight on flying stability provided by this work appears to be limited, since the same materials and meta grating pattern have been previously reported for stability behaviour. The simulator tool for evaluating the structure stability of the sail is novel, but the significance of this tool is not quite clear. For a spinning sail, the points of maximum stress are intuitive (i.e., the centre of the flat sail) and can be calculated and compared to material tensile limits without the tool. Looking at the planar silicon nitride sail and applying textbook equations for stresses near the centre of a spinning disk (radial stress = tangential stress = $(3+\nu)/8 \cdot \rho \cdot \omega^2 \cdot r^2$) gives total stress of ~222 MPa, very similar to the value from the simulator that confirms the finding that the stress is many times lower than tensile limit. The radiation pressure stresses and structural spinning stresses are quite different in magnitude in laser sails, so while the simulator can solve for both, it isn't clear that that simulator feature offers significant new insight particularly for planar sails. As the authors state, a flexible planar sail (which needs the simulator) has very similar dynamics as a rigid sail which can be modelled without the mesh-based simulator tool.

It seems the tool can be useful to identify failure/crumpling during acceleration for curved sails, but that seems to be of comparatively moderate importance. Given these observations on novelty and significance, I am not able to recommend publication in Nature Communications.

The beam-riding stability of curved and flat, patterned lightsails has indeed received considerable and growing attention, owing to the tremendous promise of lightsails for interstellar space exploration. Our work would not be possible if not for the achievements of prior studies. However, we fear that the reviewer does not fully appreciate the significance of our work. We argue that our manuscript represents a dramatic and critically needed leap forward for the interstellar lightsail concept, as we for the first time show that lightsails can

overcome the mostly overlooked, yet crucial remaining hurdle facing their viability as an interstellar propulsion technology: structural shape stability.

Nearly all beam-riding lightsail designs to date have assumed the lightsail to be a rigid body, with forces and torques acting about its center of mass to produce changes to position and orientation, but neglecting any changes to shape. This assumption has enabled several promising, highly optimized lightsail designs in literature, but its validity warrants scrutiny for realistic laser-driven lightsail spacecraft. A lightsail membrane will *not* be rigid if it (a) is meters in diameter but has submicron thickness, (b) has little or no structural reinforcement due to the strictly limited mass budget, (c) experiences unevenly distributed forces resulting in acceleration of thousands of g's, and (d) is made from any known materials.

The inevitable conclusion is that the multitude of rigid-body lightsail designs reported to date, if fabricated as proposed and subjected to the intended acceleration conditions for a relativistic interstellar mission, would not exhibit beam-riding stability, because within milliseconds of launch, they would collapse, rupture, tumble or veer off-course due to lack of structural stability. Our manuscript argues, by example rather than critique of prior work, that analysis of lightsails for beam-riding stability can and should be accompanied by corresponding structural modelling and simulation, and most excitingly, shows that the challenge of mechanical flexibility is not insurmountable.

To reach this conclusion, we developed a comprehensive multiphysics simulator for lightsails described in the manuscript and available online. This software is capable of modeling lightsails having both geometric shape-based and nanophotonic surface-based designs for beam-riding stability, and simulates the mechanical, optical, thermal dynamics of the membranes. Our work establishes a framework for computing the shape evolution of lightsails induced by nonuniform laser illumination and nonuniform composition, which are practical necessities for a realistic interstellar mission, and most importantly, allows one to quantify the resulting effects on structural and beam-riding stability. While shape deformations and modes of spinning membranes has been studied analytically (e.g., Delapierre et al. in *International Journal of Solids and Structures*, 2018), their interplay with and inevitable impact on beam-riding stability of lightsails necessitates finite-element numerical methods capable of modelling the dynamics of complex membrane geometries under varying spatial and temporal excitation. Despite the overt implausibility of physically rigid meter-scale interstellar lightsails, mechanical flexibility in lightsails has received almost no attention to date.

To prevent structural collapse, our work focuses on spin-stabilization at high frequencies, which, as noted by the reviewer, yields behavior similar to rigid-body dynamics, as spinning reduces shape deformations. However, simply spinning a lightsail is insufficient for achieving dynamical stability. In fact, we evaluated beam-riding designs from literature, intended for flat rigid-body lightsails, including our own; and observed incompatibility with spin-stabilization. This is because at the required spin frequencies, rotational (gyroscopic) effects disrupt the equilibrium between roll and pitch angles and the resulting restoring torques. Structures exhibiting beam-riding stability in the non-spinning rigid-body case (Ilic and Atwater, *Nature Photonics*, 2019 & Gao, Kelzenberg et. al., *ACS Photonics*, 2022), become unstable at high spin frequencies. Conversely, structures that are stable when spun (such as the paraboloid sails

in Fig. 3 and Fig. 4) are decidedly unstable without rotation. Although spinning rigid lightsails such as cones were previously analyzed (e.g., by Manchester and Loeb), the criteria for such stability has not yet been applied in the emerging field of nanophotonic membranes. Spinning-body dynamics requires Floquet theory as described in the Supplementary Information, which led us to a new self-stabilizing metagrating design for silicon nitride lightsails. This design may bear superficial resemblance to previously reported designs (Gao et al., ACS Photonics, 2022), but cannot be designed by applying previously reported techniques, and thus represent a fundamentally new and critically important step forward for the realization of passively stabilized *spinning* nanophotonic lightsails.

Furthermore, despite having identified stable metagrating designs for flat spinning lightsails through Floquet stability theory, one should not conclude that beam-riding stability is a trivial matter for flexible lightsails. Although the spin-stabilized lightsail sails appear to be essentially rigid in the figures and videos accompanying the previously submitted manuscript, this is decidedly and critically not the case, and we apologize for not making this clearer in the narrative. Their behavior is similar to, but *not* identical to the dynamics of identical rigid-body sails. In fact, we observed previous spun flexible lightsail designs to veer off course despite rigid-body based predictions of stability. Considerable effort was required to identify designs with sufficient stability margin to overcome the perturbances induced by deformations, as discussed in the Supplementary Information. Critically, it appears that flexible lightsails should be optimized not only for large restoring forces at equilibrium, but also for minimized lateral displacement of the equilibrium point over a wide range of tilt angles, as shown in Fig. 5e and Fig. 5f. This conclusion was reached only after extensive investigation of numerical simulation results, and is not a priori derivable from linearized rigid-body stability analysis for several reasons. First, even a spin frequency of 120 Hz does not produce a perfectly flat lightsail as evidenced by the simulated average angular deviations on the order of 1° and maximum angular deviations exceeding 10° (Fig. 6f and Fig. 6g). This fact is further corroborated by the visualization of out-of-plane displacement now included in Fig. S5 (see our response to reviewer 1, comment #6), and for further illustration, in the depiction of angular deviation from a flat shape as shown below.

Secondly, given these finite angular distributions in a spin-stabilized flexible lightsail and considering the limited angular range for self-restoring torques to be effective as shown in Fig. 5e and 5f, beam-riding behavior cannot be automatically conferred without simulation, especially if larger tilts are chosen as initial conditions. The optical response of nanophotonic lightsails becomes nonlinear even at relatively modest tilt angles (Fig. 5d and Fig. 5e), limiting the insights that can be obtained from linearized stability analysis. In our present work, we have studied thermal expansion and its effects on deformation and strain, and have provided the framework to implement more advanced optical models such temperature- or strain-dependent optical properties, which must be addressed if functional lightsails are to be realized. As shown in Supplementary Fig. 14 and discussed in the main text, structural flexibility *and* heating-induced strain visibly alter lightsail dynamics, but do not disrupt its beam-riding stability – another result that is challenging to conclude a priori. In fact, the thermally induced stress is also the reason for the difference between predicted stress of $((3 + \nu)/8)\rho\omega^2R^2 \approx 143.9$ MPa (with parameters $\rho = 2700$ kg/m³, $\omega = 2\pi \times 120$ Hz, $R = D/2 = 0.5$ m and $\nu = 0.27$) and simulated averaged stress of $\sim 0.0008 \times E_{\text{Si}_3\text{N}_4} \approx 216$ MPa. To make this case clearer, we

revised Supplementary Fig. 13, showing the difference in strain between the simulated lightsail subject to heat absorption and a thermally inactive lightsail (zero absorption and zero emission), which we also refer to in the main text.

All these points summarize the significance of our simulator tool for studying spinning flat lightsails, offering insights not only in verifying quasi-rigid self-stabilizing dynamics *despite* beam-induced shape deformations on the order of degrees and millimetres, but also in measuring the visible differences due to exactly those local membrane curvatures while accounting for thermal effects. Especially in recent years, growing interest in interstellar lightsail technology has led to prolific creativity and innovation in material engineering, thermal designs, and photonic concepts for beam-riding. We believe that the question is no longer whether any particular challenge can be overcome, but whether a particular design can overcome all of them at once. The time has come to transition from exciting but disjointed concepts towards practical holistic spacecraft design, and we are convinced that the tools and methods disclosed herein represent a major step in that direction.

Following our response, we have made the following revisions to the manuscript text:

For the studied duration of $t = 5$ s, the lightsail oscillates about the beam axis while remaining relatively flat and level thanks to sufficiently large tensioning forces arising from spin-stabilization. However, on closer inspection, the nonuniform beam induces shape deformations

(Supplementary Fig. 10), with a maximum displacement from a perfectly flat shape of ~ 3 mm (Supplementary Fig. 11a).

At closer look at the time-dependent pitch and roll angles reveals that the lightsail does deform (Supplementary Fig. 15), as indicated by an average angular spread of $\sim 1^\circ$ and a maximum angular spread of almost 14° (Supplementary Fig. 16). This observation emphasizes the importance of simulating flexible lightsails and verifying their self-stabilization in the presence of beam-induced angular deformations given the limited angular range for self-restoring forces and torques to be effective.

[Conclusion:]

Importantly, beam-induced shape deformations on the order of degrees and millimeters perturb, but do not disrupt the beam-riding dynamics of such flexible, meter-sized and subwavelength-thin lightsails enabled by optical metastructures.

Additional comments:

It is stated that polyimide is far too weak to span meter-scale areas between structural supports, but it is not quite clear why that would be the case. The stationary burst diameter refers to the membrane rigidly clamped at its perimeter, where the perimeter is also clamped and stationary. But the perimeter of a light sail moves with the sail, so in principle a pressure of 67 Pa could be supported for large polyimide sails, especially if uniformly applied. Authors do note that the perimeter constraint overestimates the required tensile strength, but I am not sure the suggestions provided by this metric are useful for screening materials. One possible alternative is to discuss maximum tensile stresses generated under non-uniform/gaussian pressure loads on freely moving sails.

Regrettably, our presentation of the “stationary burst diameter” figure of merit fell far short of our underlying intent in writing this section of the paper; we thank the reviewer for questioning it, and have revised the discussion significantly. First and foremost, the symbol we selected, D_{\max} , suggests an unintended implication that this is a limiting diameter for perimeter-supported lightsails. For this reason, we have changed the symbol to D_{SB} throughout the paper. We further clarify that this is a lower limit for perimeter-supported lightsail diameter in most cases. Finally, although we remain convinced that D_{SB} is the most broadly useful and conceptually tractable figure of merit to accompany this discussion, we acknowledge that the narrative would be improved by considering the case of a free-flying lightsail with perimeter support or Gaussian illumination conditions.

To provide some first-order insights into how the maximum allowable lightsail diameter might differ from the D_{SB} figure of merit, we employ the mesh simulator to analyze the specific example of a Si lightsail, for which D_{SB} was calculated to be 1.09 m. We assume a 43 nm thick (0.1 g/m^2), flat circular membrane of this diameter, attached at its perimeter to a rigid hoop. In the first case, the hoop is stationary; in the second case, the hoop is assumed to have mass equal to that of the encircled membrane (0.095 g) and is constrained in all degrees of freedom except motion along the acceleration axis, thus allowing the support hoop to accelerate with the

lightsail (note that we call this “free flying” despite the imposed constraints, which are necessary for successful simulation). As this discussion pertains to mechanical (rather than optical or thermal) properties, we assume 100% reflectance, no absorption, and no emission; such that the temperature remains at 300 K and thermal stress does not occur. The beam intensity is slowly ramped up in the time domain until tensile failure is detected, for uniform illumination and Gaussian beams of various beam waist diameters (relative to the lightsail diameter). The results are summarized below as newly added Supplementary Table 2:

Gaussian illumination conditions				Stationary rigid hoop		Free-flying rigid hoop ($m_{\text{hoop}} = m_{\text{sail}}$)		
Beam size relative to lightsail $\frac{w_0}{r_s}$	Intensity at lightsail edge $\frac{I(r_s)}{I_0}$ (%)	Beam capture efficiency (%)	Beam uniformity efficiency (%)	Rupture I_0 (GW/m ²)	Relative force vs. 10 GW/m ² uniform illumination (%)	Rupture I_0 (GW/m ²)	Rel. to 0.1 g/m ² membrane at 10 GW/m ² uniform illumination	
							Force (%)	Acceleration (%)
∞ (uniform)	100	0	100	9.1	91	17.0	170	85
2	60.7	39.3	78.6	10.3	81	17.5	140	69
$\sqrt{2}$	36.8	63.2	63.2	11.6	63	18.0	114	57
1	13.5	86.5	43.3	14.2	61	19.8	85	43
$\sqrt{2}/2$	1.8	98.2	24.6	19.0	47	23.5	58	29

Simulations assume 100% reflectance and $T = 300$ K. w_0 is the Gaussian beam waist radius. r_s is the lightsail radius. I is the beam intensity (I_0 at the center).

While the accuracy of the time-domain simulation tool is somewhat limited compared to analytic solutions or finite-element solvers, it is adequate for this illustration. As summarized in the table, assuming either finite Gaussian (versus uniform) illumination, or a finite-mass free flying (versus stationary) perimeter support, permits the peak incident power intensity I_0 to exceed 10 GW/m² without rupturing the membrane. Thus, the membrane could be made larger, or operated under higher peak power intensities, than suggested by the D_{SB} figure of merit.

These cursory simulations yield the burst intensity for a lightsail of fixed diameter D_{SB} , rather than the burst diameter for the fixed I_0 value of 10 GW/m² established at the outset of the discussion. Although we do not quantify specific burst diameter values at this intensity for comparison to D_{SB} , it follows that a higher burst intensity for a certain lightsail diameter should equate to a larger burst diameter at a lower intensity. Ultimately, the maximal propulsion intensity will likely be limited by the components and materials of the lightsail spacecraft, in which case a near-uniform beam of this intensity would produce maximal acceleration. However, achieving high beam uniformity across the entire lightsail area at the diffraction limit would require a dramatically larger, more costly, and less efficient laser propulsion system, and further makes passive beam-riding more difficult. These are trade-offs that must be optimized at the systems level, where structural stability of the lightsail is one of many critical requirements.

We further regret making the unsubstantiated claim that conventional solar sail materials (polyimide) are unsuited for this application based on mechanical properties alone, given that our analysis provides only a lower limit rather than upper limit for spannable membrane area. We do believe that such materials are unsuitable for interstellar lightsails, but this is based on a multitude of unstated factors in addition to mechanical strength, and we thank the reviewer for calling out this gap in logic, as previously presented. This issue is ultimately tangential to the primary focus of our paper; thus we have elected to omit comparative statements regarding conventional solar sail materials, rather than expand upon the reasons why we think these materials are poorly suited for interstellar lightsails.

The design and analysis of specific (“realistic”) support structures is a topic of great interest, but to treat it with similar rigor as we devoted to the spin-stabilized lightsails is beyond the scope of the present manuscript. Our goal in presenting this figure of merit is to establish that it may be possible or perhaps promising to design perimeter-supported lightsails. Although we can offer quite limited practical insight within such a brief discussion, we do subsequently present simulation methods (and publicly available code) that are specifically suited to address the performance of a wide variety of lightsail designs and support schemes, which we believe will be useful to the community in future efforts to develop and improve the understanding of externally supported lightsails. We have included this response to the revised Supplementary Information as Supplementary Note 2, and revised the manuscript text accordingly:

This pertains to the construction of a perimeter-supported lightsail, e.g., spanning a ring-shaped support frame, but rather than making specific assumptions about the mass, rigidity or pretensioning of such a support structure, we consider the simpler and more conservative case in which the perimeter is stationary and rigidly clamped without pretensioning. Precluding free flight of the membrane, D_{SB} should thus be interpreted as a comparative figure of merit rather than a practical size limit for perimeter-supported interstellar lightsails. Realistically, the support structure must have finite (preferably small) mass so that it could be accelerated with the lightsail, and the beam would necessarily taper off at the lightsail edge, enabling larger lightsails to be constructed than indicated by D_{SB} (see Supplementary Note 2 for example cases). The design of a practical lightsail spacecraft must address its specific support structure(s) and payload(s), and must also consider optical and mechanical properties of the membrane throughout the range of illumination conditions and operating temperatures experienced during acceleration – none of which are captured by D_{SB} , although we address some of these issues in greater detail in our numerical simulations below.

Curved sails in Figs. 3 & 4 have the added complexity that there are multiple reflections for radiation thrust and also thermal radiation (the inner surface radiation configuration factor is not zero). The latter would lead to additional heating. It is not clear if the temperature calculations account for that. If not, there is possibly a simple way to estimate the temperature discrepancy (to first order, relative scaling with a configuration factor could account for the fact that parts of the sail emit/receive thermal radiation into/from other parts of the sail and not just space)

The simulations include the localized heating resulting from absorption of reflected laser illumination, as evidenced by the “hot spots” in the temperature map images, but do not include

radiative heat transfer from one region of the membrane to another. The reviewer is correct in noting that the simulations thus likely somewhat underestimate the temperature of deeply curved lightsail membranes.

Applying this radiative heat transfer analysis to the paraboloid Si lightsails from Fig. 3 and Fig. 4 yields several insights. First, the magnitude of radiative self-heating is somewhat negligible compared to the magnitude of absorptive heating and radiative cooling, especially that resulting from the focused secondary reflections of the incident beam, which dramatically increase the local temperature. Supplementary Fig. 5 shows the results of radiative heat transfer simulations for an example scenario in which the secondary reflections lead to a peak temperature of ~ 650 K. Adding radiative heat transfer to the model increased the average sail temperature by up to ~ 4 K, although the peak maximum temperature was essentially unchanged (in a simulation without secondary reflections, the average temperature increase due to radiative heat transfer was ~ 2 K at the same simulation time). Note that the simulation shown in the figure below (also available as Supplementary Fig. 5) differs slightly from that presented in Fig. 4 (primarily in that we chose a lower spin frequency [90 Hz], which destabilized beam-riding). We found the effects of RHT to be similar across other simulations, including those with prolonged steady-state beam-riding (≥ 1 s). For example, the 100 Hz spin-stabilized lightsail shown in Fig. 3 exhibited a steady-state 2.6 K increase in average temperature, and a 1.7 K increase in peak temperature, when we enabled radiative heat transfer in the model.

Supplementary Figure 5. Comparison of simulation results with and without the consideration of radiative heat transfer (RHT), for a Si paraboloid lightsail of the same geometry as presented in Fig. 3 and Fig. 4 (some conditions differ; see text). (a – d) show lightsail temperatures versus time. (e – j) show the spatial distribution of relevant values throughout the mesh at $t = 36$ ms.

Thus, radiative heat transfer appears to have little effect on the lightsail temperature, due in part to the relatively low assumed emissivity of 0.1, meaning that only 10% of the self-radiation is reabsorbed. Radiative heat transfer could be more prominent for lightsails with high emissivity, or geometries with higher view factors.

More details of the calculation and implementation are described in Supplementary Note 4.

Figure 3-top-right: it can be observed that a paraboloid sail with spin 90 Hz remains bounded for longer than flat sail (blue, red) or paraboloid sail (orange, non-spinning). Does this observation depend on the chosen starting conditions? It would be helpful to discuss if/how starting conditions affect these observations.

The simulated collapse and beam escape for the unstable lightsail shapes is indeed sensitive to the chosen starting conditions. There are several starting variable perturbations that can impact perceived stability, including, but not limited to sail diameter, thickness, material, height/aspect ratio, shape, and surface optical models; beam profile, diameter, power, and time-variation; initial spin frequency and initial lightsail-beam misalignment in position and angle. Their interactions can be somewhat complex, especially for spinning structures, owing to the precession, nutation, and time-varying deformation. With our time-domain simulator, we cannot guarantee absolute (indefinite) stability of flight; we can show only that the trajectory remains bounded over the finite duration of the simulation time. We currently do not have the computational resources necessary to run many high-fidelity long-duration simulations, which would be needed to construct reasonably detailed ‘stability region diagrams’ to elucidate the underlying stability trends with the time-domain simulator. However, with further computational power and effort, it should be possible to perform extensive parameter sweeps or employ modern optimization or machine learning algorithms to extend the predictive or generative capabilities of time-domain simulations.

At present, we believe that the mesh simulator is better suited to understand and optimize specific designs and acceleration scenarios, whereas obtaining a broader parametric understanding of the underlying stability criteria is better achieved through analytical calculations, such as the rigid-body Floquet analysis that drove our design of the flat metasurface lightsails in the present manuscript. That linearized stability analysis approach was successfully employed to assist our design process in the case of small deformations due to spin stabilization, but cannot be applied to flexible structures with substantial deformations over time. This confirms the need to employ time-domain methods to validate and optimize the design in the flexible case.

We note that our simulation code can perform instantaneous analysis of forces and torques for arbitrarily shaped sails using rigid-body dynamics, which can be used for Floquet stability analysis. This would be an interesting approach to analyzing stability criteria for highly spin-stabilized structures that exhibit negligible deformation, and would leverage the rather

extensive optics and physics models implemented in the present code. However, this effort would fall outside the scope of the present manuscript, which, for curved lightsails, focuses primarily on presenting novel simulation techniques to model the destabilizing effects of shape distortion rather than identifying specific criteria for stability.

Finally, by expanding the set of basis state vectors, one could capture the dynamics of the primary deformation modes, or, in principle, all degrees of freedom in the underlying mesh. It is beyond our expertise to predict whether linearized analysis would be adequate to predict stability in this case, wherein advanced analytical techniques could be leveraged to gain more predictive insights into lightsail design stability. The scope of our present manuscript is limited to illustrating, through specific examples, the novel time-domain simulation techniques that were developed while studying our self-stabilizing flexible lightsails. In conjunction with publicly releasing the simulation code to the public, we hope to enable others to investigate other specific designs and parameters of relevance to their research interests. Thus we have added the following statement to the manuscript:

The initial beam-lightsail alignments were specifically chosen for the presented exemplary acceleration scenarios, with dynamical stability being highly sensitive to those starting conditions.

The last sentence in the abstract that refers to the lightsails and their fabricability (“...and that their [membrane-like lightsails] fabrication should be within the reach of modern microfabrication technology”) is not quite correct. The work shows nano-element features that have dimensions that can be fabricated. But that does not support the claim that light sails of needed meter-scale and sub-micron thickness are, at present, within the reach of fabrication technology. Clarifying the difference between the two would be helpful.

We thank the reviewer for this constructive criticism. In our work, we show that self-stabilization of spinning flexible lightsails can be achieved not only via shape engineering, but also via nanophotonic structuring of initially flat surfaces. We find the latter particularly encouraging because modern microfabrication technology can already produce flat nanophotonic structures at the wafer scale. Therefore, we expect that lightsail scale-up efforts could benefit from the decades of expertise and immense industrial scale of the semiconductor and photonics wafer fabrication industries. We did not intend to suggest that lightsails can be fabricated at the meter scale with present-day technology. For this reason, we have revised the last sentence in the abstract as follows:

These advances suggest that laser-driven acceleration of membrane-like lightsails to the relativistic speeds needed to access interstellar distances is conceptually feasible, and that their fabrication could be achieved by scaling up modern microfabrication technology.

In Fig. 6f,g, the scalebar uses “Norm(Log(Max(Counts,10/10))” units. It would be helpful to describe the scale bar units either in the caption or in the main text closer to where the figure is referenced.

We thank the reviewer for the helpful suggestion. In addition to adding a short explanation of the scale bar to the caption of Fig. 6, we have also revised Fig. 6f and Fig. 6g by considering plotting the angle distributions for bins of width 0.1° with at least one instead of with at least ten elements as done previously. We hope that this does not only simplify the interpretation of scale bar, which now reads “Normalized $\log(\max(\text{counts},1))$ ”, but will also highlight the angular distribution within the lightsail and thus its flexible nature more clearly.

(f), (g) Distribution of θ and ϕ angles, respectively, of mesh elements of the flexible lightsail versus time, with the color bar depicting normalized counts on a logarithmic scale with a bin width of 0.1° , [...]

Reviewer #3 (Remarks to the Author):

In this work, the authors present a detailed calculation of the motion of laser-propelled lightsails, and demonstrate many sail configurations that exhibit stable motion, where the sail remains within the drive laser beam, and does not crumple or decompose. This is a very complete study, which considers the thermal and mechanical properties of the constituent materials, multiple sail geometries, and - for the first time that I am aware of - the potential bending motion and internal tension of the sail under acceleration. The consideration of these latter two parameters make this work exemplary and very interesting to a broad community of scientists who either work directly with laser sails, or have looked on with interest wondering if they are at all feasible. By explicitly including bending and potential fissuring of a flexible sail, the authors have shown unequivocally that stable laser sails are possible, even without a rigid structure, and for very thin membranes. As such, I fully endorse this work for publication in Nature Communications.

We thank the referee for the positive evaluation and endorsement of our work and for highlighting the significance of our results.

I do, however, feel this manuscript could be improved by a few inclusions and explanations:

1 - The authors explicitly state that bending stiffness should be 'negligible', and is not considered in their 2D mesh geometry, but it is not immediately obvious to me that this is the case. I would naively expect that bending stress would be a bigger problem than in-plane tension. Could the author's include a rough estimate of what the bending stress is for the flexible, spinning SiN membrane, described in Fig. 6 (the seemingly most promising design)? I understand that full scale simulations are beyond the scope, but it would be insightful if they could make a rough calculation that considers the sail curvature and local forces at a particular point in time.

For the simulations of flexible lightsails reported in this work, bending stiffness can be considered negligible due to their extreme aspect ratio (meter-scale, but submicron thin) and high spinning frequencies. If bending stresses are accounted for in our lightsails, then those will be significantly smaller than the in-plane stresses induced by spinning the lightsail. To show this, we use the Föppl-von Kármán equations to describe the (normal) deflections $w(\mathbf{r})$ of a thin elastic disk given by

$$D\nabla^4 w = h\nabla \cdot (\mathbf{T} \cdot \nabla w),$$

Where the left-hand side characterizes bending of the disk determined by its flexural rigidity D , whereas the right-hand side governs in-plane or stretching related behavior governed by the in-plane stress tensor \mathbf{T} , which depends on the applied body force per unit mass \mathbf{b} via

$$\nabla \cdot \mathbf{T} = -\rho\mathbf{b}.$$

For spinning lightsails, \mathbf{b} can be expressed in terms of the angular velocity ω , assuming zero angular acceleration, in the rotating frame of the disk:

$$\mathbf{b} = r\omega^2\hat{\mathbf{r}}.$$

The natural length scale of normal displacement is given by the disk radius R , resulting in

$$\nabla \cdot \mathbf{T} = -\rho\mathbf{b} = -\rho R\omega^2\hat{\mathbf{r}} \rightarrow \mathbf{T} \sim \rho R^2\omega^2,$$

Due to the gradient scaling as $1/R$. Therefore, the right-hand side of the Föppl-von Kármán equation proportional to in-plane stresses scales as

$$\sigma_{\text{in-plane}} \sim h \frac{1}{R^2} (\rho R^2 \omega^2) = h\rho\omega^2,$$

Compared to bending stress (left-hand side of the Föppl-von Kármán equation) scaling as

$$\sigma_{\text{bending}} \sim \frac{D}{R^4},$$

Consequently, the ratio of in-plane stress induced by spinning to bending stress for a thin disk (neglecting shear effects) can be calculated as

$$\frac{\sigma_{\text{in-plane}}}{\sigma_{\text{bending}}} \sim \frac{h\rho\omega^2}{D/R^4} = \frac{\rho h\omega^2 R^4}{Eh^3/12} = \frac{12\rho\omega^2 R^4}{Eh^2},$$

Inserting numbers according to the simulation reported in Fig. 6, with $\rho = 2700 \text{ kg/m}^3$, $E = 270 \text{ GPa}$, $h = 400 \text{ nm}$, $R = 0.5 \text{ m}$ and $\omega = 2\pi \times 120 \text{ Hz}$, we find

$$\frac{\sigma_{\text{in-plane}}}{\sigma_{\text{bending}}} \sim \frac{12\rho\omega^2 R^4}{Eh^2} \approx 4.26 \times 10^{11},$$

Proving that bending stiffness can indeed be considered negligible in our simulations due to the use of spin-stabilization for the ultrathin, meter-scale flexible lightsails. We have revised our statement regarding bending stiffness as follows:

Due to the extreme aspect ratios of the lightsail membranes, having meter-scale extent and submicron thickness, bending stress is negligible compared to the in-plane stresses induced by the chosen spin frequencies, and bending stiffness is negligible compared to the out-of-plane forces induced by optical propulsion and membrane elastic deformation. Therefore, bending stiffness is omitted from the model, and the specific shear modulus of the material is also neglected.

2 - In Ref. 21, spinning conical sails are considered and the author of that work explicitly states that if the axis of angular momentum is not aligned with the laser beam, the sail would be unstable. That work also provides a brief explanation describing why that is the case. In this work, the authors do explicitly mention the problem described in Ref. 21, but it is unclear how it is resolved in this work - i.e. why the spinning cases seem to avoid destabilization. Can the authors provide some explanation?

In our simulations of spinning curved lightsails (only paraboloids are shown in the manuscript, but we have also studied cones, spherical sections, and other shapes, some of which are depicted in Supplementary Video 2), we did not assume initial angular misalignment of the lightsail to the beam axis. Only the metagrating-based (flat) lightsail was subject to angular misalignment, in which case we confirm bounded (stable) trajectories, for modest initial misalignments. For the curved lightsails, the spin axis is initially parallel to the beam axis, which, for sufficiently high spin frequencies, allows perturbing lateral forces to be averaged out over the precession period as described by Manchester and Loeb (The Astrophysical Journal Letters, 2017). Therefore, our results are consistent with prior works.

More generally, we find this an intriguing question worthy of further investigation, as alluded to by its brief mention in the manuscript. Our simulation tools are well suited to model spinning curved sails as either flexible membranes or rigid bodies. We were tempted to and in fact did start performing of tilt misalignments for other geometries. Our initial conclusion is that the matter should be more thoroughly investigated and published separately, rather than presented as a cursory addition to the supporting information in the present manuscript.

To clarify this, we have revised the manuscript text as follows:

With adequate spin stabilization ($f_{\text{spin}} = 135$ Hz), the shape remains stable, and beam-riding stability is achieved throughout the simulated 1 s duration due to destabilizing forces being averaged out over a full precession period without an initial tilt²¹

3 - The precise method for modeling the motion of the sail is not completely clear. The authors state they model the dynamics in timesteps that are 10 - 20x shorter than the smallest periodic motion, but otherwise it is not clear what exactly is being done. Are they using a Runge-Kutta method?

We thank the reviewer for noticing the lack of clarifying the precise method for numerical integration of the flexible lightsail's equations of motion. All reported results in the submitted manuscript were obtained according to the following equations

$$\mathbf{v}_n(t_i + \Delta t) = \mathbf{v}_n(t_i) + \frac{\mathbf{F}_n(t_i)}{m_n} \Delta t, \quad \mathbf{r}_n(t_i + \Delta t) = \mathbf{r}_n(t_i) + \mathbf{v}_n(t_i) \Delta t,$$

Where n corresponds to the i -th node of the 2D-mesh modelling our lightsails. This time stepping approach corresponds to the symplectic (semi-implicit) Euler method, which in contrast to the Runge-Kutta is a semi-explicit (or semi-implicit) method and thus benefits energy conservation (albeit energy is only monitored and not further processed in our simulations).

Since submission, we have also implemented the option to use the Runge-Kutta method for numerical integration in the simulator code, which may be more accurate for longer-duration simulations. In the figure below, we compare the simulated translations along x and y of the reported flexible silicon nitride lightsail using the Runge-Kutta method (RK 4) to our original

results based on the semi-implicit (SI) Euler method reported in Fig. 6c in terms of the maximum mean change (left y-axis) and absolute difference (right y-axis).

We observe that during the first three seconds of acceleration, the maximum mean change is less than 2% between the results based on the symplectic Euler method and RK. Only after four seconds does the error start to grow to more than 6%.

For this reason, we performed new simulations and have replaced all results shown in Fig. 6 with new data based on the Runge Kutta method simulations, while choosing to keep the results based on the semi-implicit Euler method for Fig. 3 and Fig. 4 due to their shorter simulated acceleration. In general, we have compared various simulations with both methods, confirming that they produce overwhelmingly similar results in terms of shape evolution and trajectory stability. Selection of the integration method is available to the user in the online version of the code. Moreover, we have included a short description of the time-stepping method in the Methods section, and more details to the numerical integration methods as Supplementary Note 3. The manuscript text was revised to include:

Specifically, the dynamics of lightsails reported in Fig. 3 and Fig. 4 were obtained using the symplectic Euler method, whereas the acceleration of the flat flexible lightsail presented in Fig. 5 and Fig. 6 was simulated via numerical integration based on the Runge-Kutta method.

4 - Many papers in this field parameterize sail performance in terms of acceleration distance. Could the authors calculate the acceleration distance for their sails shown in Fig. 6 and include those numbers in the manuscript?

To calculate the exact acceleration distance to $0.2c$ for the lightsail design shown in Fig. 6, we would need to run the time-domain simulations for the entire acceleration phase, which due to reasons of computational resources and numerical inaccuracies is presently intractable. At the end of the simulated duration of 5 s, the flat flexible spinning silicon nitride lightsail is accelerated to a velocity of ~ 1.86 km/s, travelling a distance of ~ 4.65 km.

If we extrapolate these numbers to a final velocity of $0.2c$, the acceleration distance will be ~ 6.6 Tm for the assumed laser peak intensity of 1 GW/m² - considerably greater than the Starshot interstellar mission target, and falling short in terms of performance compared to other reports (see our response to reviewer #1, comment 9). While focusing on simulation of higher performing lightsail designs is an important future goal, we have focused this manuscript on

introducing new methods and techniques to achieve dynamic stability in *flexible* lightsail membranes, which has not yet been reported in the literature.

Motivated by our parallel experimental efforts, we decided to constrain our designs to the silicon nitride material system, and to geometries within reach of laboratory fabrication equipment. This decision precluded the consideration of higher-index materials such as MoS₂, which also show promise to dramatically increase the reflectance of lightsails (Brewer et al., Nano Letters, 2022) and could likely also reduce the areal mass density required to achieve a self-stabilizing metagrating. Furthermore, we assumed and calculated absorption and emissivity for silicon nitride based upon literature or experimental data. Higher emissivity could eventually be achieved through future photonic engineering as reported by Brewer et al., or absorption could be reduced through future improvements in fabrication, e.g., during deposition or thermal annealing. Consequently, this restricted our design to a laser intensity $\sim 10\times$ lower than that targeted for the canonical Starshot interstellar mission to avoid overheating, required the membrane to be thicker than desired to achieve beam-riding stability, and resulted in relatively low effective reflectance of $\sim 10\%$. Nevertheless, the simulated design represents a lightsail, which can be currently fabricated and tested in the lab, and possibly scaled up.

Finally, we note that due to the expected Doppler shift at higher velocities, the optical behavior of our metagratings would differ at later stages of acceleration, to the point where they may not necessarily retain their beam-riding characteristics. Modelling full acceleration dynamics will require further improvements to the simulator code for more robust and efficient calculations, in addition to optical and mechanical characterization of materials over wider wavelength ranges and operating temperatures, which is beyond the scope of this manuscript. Nonetheless, we believe that our code and the insights drawn from our results will provide the basis for such improvement and optimization to achieve self-stabilizing, flat flexible lightsails made of silicon nitride for relativistic velocities. We have included a short discussion of this in Supplementary Note 14.

REVIEWER COMMENTS

Reviewer #1 (Remarks to the Author):

The authors have also addressed all my previous comments and I am happy to recommend the paper for publication. The paper significantly advances the field of deep-space lightsails by performing pioneering time-dependent optomechanical simulations and identifying designs of lightsails that can maintain their shape under acceleration and ride the beam in a passively stable manner.

Reviewer #2 (Remarks to the Author):

I appreciate the comments by the authors though I still harbour concern about the impact of the provided insight for publication in Nature Communications. The simulator is a valuable tool for curved sails, but curved sails have already been recognized to be significantly less practical than planar sails. For planar sails, the simulator is helpful for flexible sails spun at lower speeds where the shape is not rigid, but that is less the case for rigid-like flexible sails where the behaviour is very similar to that of a fully rigid sail. Regarding the stress comparison, the authors noted “the thermally induced stress is also the reason for the difference between predicted stress of $((3 + \nu)/8)\rho \omega^2 R^2 \approx 143.9$ MPa (with parameters $\rho = 2700$ kg/m³, $\omega = 2\pi \times 120$ Hz, $R = D/2 = 0.5$ m and $\nu = 0.27$) and simulated averaged stress of $\sim 0.0008 \times E_{\text{Si3N4}} \approx 216$ MPa.” Inserting these into the equation for radial stress = tangential stress = $(3 + \nu)/8 \rho \omega^2 R^2 \sim 156.8$ MPa, wouldn't the total stress near the sail centre be $\sqrt{\text{radial_stress}^2 + \text{tangential_stress}^2} \sim 221$ MPa which is very similar to the simulated averaged stress? It would be helpful if clarification can be provided.

That being so, can it be said that this spinning sail is actually stable?

Stability requires a restoration mechanism for perturbations along degrees of freedom accessible to the object, but it is not clear how the sail stabilizes for perturbations about the z-axis (rotations about the z axis).

If I understand correctly, the assumption in this work is that the laser also “spins” in synchrony with the spinning sail at the same frequency and in phase. This is required to keep red/blue patterns in Fig. 5 aligned with laser polarisation: red parts are always exactly perpendicular to E-field and blue parts are exactly parallel to E-field.

If a sail is rotationally perturbed about the x-axis or y-axis, there will be a restoring torque. But what would happen if the sail is similarly rotationally perturbed about the z-axis?

The nano-patterned sail does not have rotation symmetry, so perturbations about the z-axis could lead to a torque and affect the balance between restoring forces F_x , F_y , etc. If those forces and torques are not stabilizing it would lead to faster/slower spinning. Behaviour that is not stabilizing along the z-axis will then affect stability along other axes because the blue/red parts are no longer aligned with the polarisation.

I understood that along the z-axis the sail accelerates so restoring z-force is not needed. But the behaviour about the z-axis would need to be stabilizing to support the notion that this sail is dynamically stable.

Summary of authors' response to reviewers

We would like to thank the reviewers for their continuous careful assessment of our work and for their constructive comments. In this response letter, we address every comment and concern of the referees and highlight the resulting changes to the manuscript and supplementary information accordingly. We summarize high-level changes on the first page, then provide detailed point-by-point responses to each reviewer comment, formatting our responses in blue for ease of reading. We also quote specific passages from the manuscript that were changed or added in response to the reviewers' comments, highlighted below in yellow. The modified manuscript has been resubmitted as a marked-up version showing all changes.

Manuscript text

- a) We have revised the introduction to elaborate more on the current state of interstellar exploration in the context of exoplanet research and space probes that have reached interstellar space, for the general audience of *Nature Communications*.

Supplementary information

- a) In response to Reviewer 2, we added a new Supplementary Note 8, in which we present an alternative metagrating design to stabilize the lightsail against angular perturbations about the z -axis in addition to the x - and y -axis. The emergence of a yaw-restoring torque, along with minimal changes to the other restoring torques and forces, is shown in newly generated Supplementary Fig. 9.
- b) To better highlight the capabilities of our flexible lightsail simulator following concerns of novelty raised by Reviewer 2, we performed new simulations comparing flexible and rigid metagrating-based lightsail dynamics at lower spin frequencies. These new results highlighting the difference between flexible- and rigid-body lightsail dynamics more clearly are discussed in a new Supplementary Note 15 and are summarized in Supplementary Fig. 19.

REVIEWER COMMENTS

Reviewer #1 (Remarks to the Author):

The authors have also addressed all my previous comments and I am happy to recommend the paper for publication. The paper significantly advances the field of deep-space lightsails by performing pioneering time-dependent optomechanical simulations and identifying designs of lightsails that can maintain their shape under acceleration and ride the beam in a passively stable manner.

We thank the referee for the recommendation for publication and the supportive assessment of the achievements and impact of our work.

Reviewer #2 (Remarks to the Author):

I appreciate the comments by the authors though I still harbour concern about the impact of the provided insight for publication in Nature Communications. The simulator is a valuable tool for curved sails, but curved sails have already been recognized to be significantly less practical than planar sails. For planar sails, the simulator is helpful for flexible sails spun at lower speeds where the shape is not rigid, but that is less the case for rigid-like flexible sails where the behaviour is very similar to that of a fully rigid sail.

We thank referee 2 for the critique of the work, which have led to further improvements to the manuscript. We wholeheartedly agree with the notion that flat lightsails are likely to be much more practical than curved sails, but felt it was prudent to consider both, given that the majority of prior reported lightsail stability studies discuss curved shapes. We hope that the novel observations we present on the topic of curved lightsails, such as the fact that surface curvature alone is inadequate to resist shape collapse, and the potentially destabilizing effects of multiply reflected light, will help guide the field towards more practical lightsail designs.

Regardless of lightsail shape, prior stability studies have assumed the lightsail to be a rigid body. This is an extremely unrealistic assumption for large, thin, ultralight membranes, which undermines the practical utility of any results that require this assumption. One of the most pressing questions in the lightsail community is that of whether dynamical stability can also occur for flexible lightsails. By treating the lightsail as a flexible membrane, then developing techniques to stabilize its shape and trajectory, we can report, for the first time, a lightsail design which exhibits both shape stability and beam-riding stability. This is an important conclusion and a substantial step forward for the field of lightsail research, as it begins to transition from theoretical work to practical demonstrations.

We focused on comparing trajectories of flexible and rigid lightsails spinning at high frequencies, which are broadly similar, but nonetheless different in detail due to the effects of mm-scale shape deformations and thermal expansion. However, our simulator tool allows to compare flexible- and rigid-body dynamics of flat lightsails at *any* spin frequency. In particular, at lower spin frequencies, the amount of mechanical flexibility will increase, thus reducing the degree of similarity between rigid and flexible trajectories. To demonstrate this, we performed new simulations assuming an initial beam-lightsail misalignment of $x_0 = y_0 = 1$ cm and $\theta_0 = \phi_0 = -1^\circ$. For a spin frequency of 75 Hz, after two seconds of acceleration, both flexible and rigid lightsail remain on a bounded trajectory. However, they travel along visibly different paths in the xy -plane, as compared to the case of 120 Hz presented in Fig. 6. At a spin frequency of 60 Hz, even though both lightsails eventually escape the laser beam, they do so at different times. Specifically, the rigid lightsail spirals out of the laser beam immediately, reaching an arbitrarily chosen circumference at $r = 0.2$ m after 0.449 s. On the other hand, the flexible lightsail manages to stay in vicinity of the beam more than twice as long, encircling the beam center more than once before veering off-course and reaching $r = 0.2$ m at $t = 1.179$ s. Finally, if both lightsails are spinning at 71 Hz, the rigid lightsail flies away, while its flexible counterpart traverses a bounded trajectory, thus exhibiting self-stabilizing behavior throughout the entire simulated flight duration of 2 s. This observation not only highlights the importance and

capabilities of our flexible lightsail simulator, but also suggests that structural flexibility in spinning lightsails indeed could benefit self-stabilizing dynamics in certain cases. Longer simulations and more studies with many distinct initial conditions as well as a closer look at

the role of mechanical deformations and modes will be needed to draw a definitive conclusion on this point.

We now refer to these new results in the revised manuscript text with the following sentence:

Specifically, at lower spin frequencies, differences between flexible- and rigid-body dynamics become more apparent, culminating in a situation where a rigid lightsail veers away from the beam, while its flexible version remains on a bounded trajectory (Supplementary Fig. 19), highlighting the importance of modeling mechanical deformations and inferring potential benefits for self-stabilizing lightsail acceleration.

Moreover, we have included a discussion of this point and an accompanying new figure as a newly added Supplementary Note 15 and Supporting Fig. 19.

Regarding the stress comparison, the authors noted “the thermally induced stress is also the reason for the difference between predicted stress of $((3 + \nu)/8)\rho \omega^2 R^2 \approx 143.9$ MPa (with parameters $\rho = 2700$ kg/m³, $\omega = 2\pi \times 120$ Hz, $R = D/2 = 0.5$ m and $\nu = 0.27$) and simulated averaged stress of $\sim 0.0008 \times E_{\text{Si3N4}} \approx 216$ MPa.” Inserting these into the equation for radial stress = tangential stress = $(3 + \nu)/8\rho \omega^2 R^2 \sim 156.8$ MPa, wouldn't the total stress near the sail centre be $\sqrt{\text{radial_stress}^2 + \text{tangential_stress}^2} \sim 221$ MPa which is very similar to the simulated averaged stress? It would be helpful if clarification can be provided.

We apologize for the confusion on this matter; indeed, the reviewer is correct in writing that the total stress at the center of the lightsail should be the square root of the sum of radial stress squared and tangential stress squared. In our case, this means that, neglecting shear effects, the theoretically predicted total stress is ~ 204 MPa.

We would like to further clarify our initial argument of the role of thermal stress in our simulations. In the manuscript, we state that the “simulation predicts a maximum strain of 0.091% in the Si₃N₄ membrane (Supplementary Fig. 14a), which translates to a tensile stress of ~246 MPa”. This simulated maximum stress is a result of both mechanical and thermal stress: the former being due to spinning the lightsail at 120 Hz, the latter due to thermal expansion arising from an absorption-induced temperature rise. To distinguish between these two effects, we ran a simulation of a thermally inactive (zero absorption and zero emission) flexible lightsail, for which we evaluated a maximum stress of ~ 182 MPa (see Supplementary Fig. 14a). Therefore, we can conclude that accounting for thermal stress increases the maximum stress in our spinning flexible lightsail by 26%. Such a change is not captured by the equation of radial and tensile stresses for a spinning disk, and thus highlights the capability of our simulator to simulate multiphysics-induced stress.

We should note that the ~10% difference between the simulated ~ 182 MPa stress for a thermally inactive flexible lightsail and the theoretically predicted ~ 204 MPa stress arises primarily due to mesh discretization. Specifically, our mesh geometry cannot sample stress exactly at the center of the disc, where it is the highest. The center is always the location of a mesh node, whereas stress is computed over mesh edges. This and inherent limitations of the mass-spring model compared to more advanced techniques in finite-element modelling, produce a slightly lower peak stress values than theory predicts. We typically employ a mesh comprising 5400 triangles to simulate lightsails in our model, which is adequate to capture shape deformations. We were able to verify that upon increasing the number of mesh elements by a factor of 25, a simulated stress of ~ 200 MPa is found, agreeing more closely with theory. However, we found that using such a fine mesh resolution is currently not computationally tractable at present for performing seconds-long dynamic simulations of large, flexible lightsails, with our available computational resources.

We have revised Supplementary Note 11 to note this point, adding this text:

We note that due to the effects of mesh discretization and chosen resolution given available computational resources, reported peak strain and thus stress values are being underestimated by ~ 10%.

That being so, can it be said that this spinning sail is actually stable? Stability requires a restoration mechanism for perturbations along degrees of freedom accessible to the object, but it is not clear how the sail stabilizes for perturbations about the z-axis (rotations about the z axis).

If I understand correctly, the assumption in this work is that the laser also “spins” in synchrony with the spinning sail at the same frequency and in phase. This is required to keep red/blue patterns in Fig. 5 aligned with laser polarisation: red parts are always exactly perpendicular to E-field and blue parts are exactly parallel to E-field.

If a sail is rotationally perturbed about the x-axis or y-axis, there will be a restoring torque. But what would happen if the sail is similarly rotationally perturbed about the z-axis?

The nano-patterned sail does not have rotation symmetry, so perturbations about the z -axis could lead to a torque and affect the balance between restoring forces F_x , F_y , etc. If those forces and torques are not stabilizing it would lead to faster/slower spinning. Behaviour that is not stabilizing along the z -axis will then affect stability along other axes because the blue/red parts are no longer aligned with the polarisation.

I understood that along the z -axis the sail accelerates so restoring z -force is not needed. But the behaviour about the z -axis would need to be stabilizing to support the notion that this sail is dynamically stable.

We thank the reviewer for raising this important point. In our work, the dynamics of flat spinning flexible lightsails patterned with optical metagratings assume perfect yaw-angle alignment between the electric field of the incident laser beam and the orientation of the metagratings throughout the duration of acceleration. This corresponds to a condition in which the angular rotation frequency of a linear polarized laser source matches the angular frequency of the lightsail, which could be done for a phased array laser source. Consequently, rotational perturbations about the z -axis cannot occur, and we base our observation of dynamical stability on the restoring in-plane forces and torques about the x - and y -axis. We made this simplifying assumption as a matter of practicality, as it reduced the complexity of computations, which allowed us to analyze and optimize the metagratings using Floquet theory, and to simulate reasonable acceleration durations to draw conclusions of dynamical stability in the transverse plane. In particular, we neglected any dependence of the optical pressures on the yaw angle ψ , and computed the optical properties of each metagrating under a single polarization, over a 2D sweep of incidence angles.

Modelling comprehensively the optical dependence on yaw angle would be useful and is something we plan to incorporate in future refinements of the lightsail simulator; however, it is beyond the scope of the current manuscript. We chose to focus on transverse beam-riding and tilt stability about the x - and y -axis, as we found these degrees of freedom to be more difficult to control and thus most important for stability. We expect that future development will allow implementation of full polarization support within the simulator, which will allow for more rigorous analysis of rotational alignment stability.

Nonetheless, we acknowledge that the reviewer raises a valid question about methods to stabilize our flexible lightsail in the case of rotational perturbations about the z -axis, and so we did perform further analysis of this phenomenon. Robustness to perturbations to the alignment between the angular velocity of the laser beam and the angular velocity of the lightsail can be introduced with an altered design, in which the metagrating sections are rotated with respect to each other, as was reported by our group previously (Ilic and Atwater, Nature Photonics, 2019)

Thus, in response to reviewer 2's comments, we performed an analysis of one potential design with a chevron-shaped metagrating pattern which is shown below, in which we compare forces for the TE metagratings (colored in blue) being rotated by an angle $\delta = 5^\circ$ relative to the lightsail y -axis y_{BF} to the case without yaw rotation ($\delta = 0^\circ$). We find that such a modification barely affects the original shape and magnitude of the angle-dependent forces $F_x(\theta)$ and $F_y(\phi)$ (b). The same observation of minor changes to the self-restoring torques $\tau_x(\phi)$ and $\tau_y(\theta)$ can also be made (c), suggesting that the self-stabilization mechanism will be preserved. Importantly, we also find that a self-restoring torque about the z -axis is observed for this

modified design, allowing to potentially realign the angular velocity of the lightsail to the angular rotation velocity of the laser source upon perturbations about the z -axis.

This newly performed analysis and the figure below have been added as Supplementary Note 8 and Supplementary Figure 9, and these changes are now also referenced in the revised manuscript with the following text:

Perturbations to the beam-lightsail angular velocity alignment could potentially be addressed by a restoring yaw torque, which would be introduced by rotating the metagratings by an angle relative to the lightsail axis (Supplementary Fig. 10).

REVIEWERS' COMMENTS

Reviewer #2 (Remarks to the Author):

I appreciate the reply from the authors. Regarding the question of yaw stability, the emergence of yaw torque in Fig. S9 is promising, but it remains unclear whether this quantitatively alters the Jacobian matrix (or resulting eigenvalues) in a way that maintains stability, and, importantly, how a yaw perturbation would dynamically interact with other perturbations to affect stability. I concur with the authors that this merits further investigation. It is an important question upon which the claim of complete stability of such a sail would rest.

That said, the changes made to the manuscript have convinced me that it merits publication. I am happy to provide my recommendation.

Reviewer #2 (Remarks on code availability):

The code does not appear to have a README file or instructions for reproducing the results in the manuscript.

REVIEWER COMMENTS

In this point-by-point response, we have formatted our response in blue.

Reviewer #2 (Remarks to the Author):

I appreciate the reply from the authors. Regarding the question of yaw stability, the emergence of yaw torque in Fig. S9 is promising, but it remains unclear whether this quantitatively alters the Jacobian matrix (or resulting eigenvalues) in a way that maintains stability, and, importantly, how a yaw perturbation would dynamically interact with other perturbations to affect stability. I concur with the authors that this merits further investigation. It is an important question upon which the claim of complete stability of such a sail would rest.

That said, the changes made to the manuscript have convinced me that it merits publication. I am happy to provide my recommendation.

We thank the referee for providing a recommendation for publication, and for the valuable comments during the review process that have helped us to substantially improve our manuscript.

We fully agree with the referee that further analysis of the effects of an emerging yaw torque on the stability analysis and perturbed dynamics will be of importance for studying lightsail stability, which we believe could be done on the basis of our presented methods and results in future work.

Reviewer #2 (Remarks on code availability):

The code does not appear to have a README file or instructions for reproducing the results in the manuscript.

We thank the referee for noticing the unintended omission of instructions and/or a README file for reproducing the results in the manuscript. We have added a document to our GitHub repository with instructions on how to run the shared code to obtain the results in the manuscript.